# Bonsai Networks: Structured Pruning and Sparse Training of Foundation Models

## Abstract

The recent trend of scaling neural networks to unprecedented sizes demands efficient structured sparsity for practical deployment, yet precisely controlling sparsity levels and patterns for hardware acceleration remains a challenge. This paper introduces **A**daptive **S**oft-**Thr**esholding **A**lgorithm (ASTRA), which achieves a target sparsity by adapting the group regularization strength based on affordable sparsity characterizations. We establish ASTRA's theoretical foundations, proving the existence of stable regularizations that yield the desired sparsity. We demonstrate sublinear and linear convergence rates for both model parameters and the regularization weight in deterministic settings, and crucially, an almost sure convergence with *mean-square* rate $O(1/t)$ in the practical stochastic gradient setting. Overall, ASTRA offers a theoretically-grounded and versatile method for direct and precise control over structured sparsity, allowing pruning and fine-tuning of foundation models into Bonsai Networks: accelerator-friendly miniatures trained to match the teacher's output while preserving downstream performance.

## 1 Introduction

The groundbreaking performance of modern deep neural networks, such as Large Language Models (LLMs) and Vision Language Models (VLMs), comes at a substantial computational and memory cost, both during the training phase and at inference time (Zhou et al., 2024). While the importance of scale is an ongoing debate (Goel et al., 2025), this cost is projected to continue to increase due to their widespread adoption and the belief that further breakthroughs will require unprecedented scales. This trend has spurred extensive research on model compression, with popular methods including quantization of model weights and activations during (Jacob et al., 2018) or post-training (Frantar et al., 2023).

Another approach to model compression is network pruning, inspired primarily by foundational work such as Optimal Brain Damage (OBD) (LeCun et al., 1989) and Optimal Brain Surgeon (OBS) (Hassibi et al., 1993). A paradigm shift was spurred by the Lottery Ticket Hypothesis (LTH) (Frankle & Carbin, 2019), which demonstrated that dense networks contain sparse "winning tickets" capable of matching the performance of the dense model when trained in isolation. Although LTH validated the potential of highly sparse networks, the computational expense of finding these tickets motivated the development of more efficient sparse training paradigms.

These paradigms include Static Sparse Training (SST), where a fixed sparse topology is identified before training (Lee et al., 2019; Wang et al., 2020), and Dynamic Sparse Training (DST), which adjusts connectivity during a single training run (Evci et al., 2020; Liu et al., 2020). While prominent recent successes in pruning LLMs have come from post-training techniques such as SparseGPT (Frantar & Alistarh, 2023) and Wanda (Sun et al., 2024), the significant cost of pre-training dense models motivates more efficient DST approaches. However, a critical challenge persists for DST methods: they naturally produce unstructured sparsity where individual weights are zeroed out. Such fine-grained patterns do not translate into practical speedups on modern hardware, for which structured sparsity that prunes entire neurons or channels is essential. Yet, existing methods that promote structured sparsity within the DST paradigm often rely on complex, ad-hoc heuristics and lack a unified, theoretically grounded framework.

To address this, we approach the problem from an optimization perspective. Theoretically grounded pruning and sparse training can be viewed through two dual lenses: seeking maximal sparsity for

a given model fidelity (Aghasi et al., 2017), or minimizing training loss for a pre-specified sparsity budget. Our work focuses on the latter, which is more aligned with the practical goals of dynamic training. We propose the Adaptive Soft-Thresholding Algorithm (ASTRA), a theoretically grounded method that induces structured sparsity within a dynamic training paradigm that is applicable to stochastic gradient approaches, thus eliminating the need for dense gradient computations. Our approach dynamically adjusts a group $\ell_1$ regularization penalty, guiding iterates towards an equilibrium that satisfies a pre-specified target sparsity for desired structural patterns. Furthermore, the proposed theoretical framework provides a principled understanding of many existing heuristic-based approaches, thereby clarifying their underlying assumptions and limitations. By leveraging the soft-thresholding operator and established parameter grouping strategies (Mairal et al., 2011), ASTRA integrates seamlessly with existing online optimization methods, facilitating broad applicability and enabling hardware-aligned structured pruning.

## CONTRIBUTIONS

- We formalize a scalar root-finding view of a target sparsity with first-order oracles.
- We provide a two time-scale scheme that tracks the sparsifying regularization in $O(1/t)$.
- A group-wise extension with closed-form prox that maps to accelerator-friendly patterns.
- Our framework is a local proximal-control of several modern heuristics, opening possibilities for extensions to structured sparsity.

## 2  PRELIMINARIES: PROXIMAL GRADIENT DESCENT

We consider the task of minimizing a function $f : \mathbb{R}^n \to \mathbb{R}$ under the sparsity constraint $\|\boldsymbol{w}\|_0 \leq \kappa$ for some integer $\kappa > 0$. A common surrogate is to employ $\ell_1$ regularization:

$$\min_{\boldsymbol{w} \in \mathbb{R}^n} \left\{ F(\boldsymbol{w}; \lambda) := f(\boldsymbol{w}) + \lambda \|\boldsymbol{w}\|_1 \right\}, \tag{1}$$

where $f : \mathbb{R}^n \to \mathbb{R}$ is convex, $L$-smooth, and $\mu$-strongly convex and $\lambda > 0$ is the regularization weight. Let $\partial \|\cdot\|_1$ denote the subdifferential of the $\ell_1$-norm; an element $\boldsymbol{g} \in \partial \|\boldsymbol{w}\|_1$ satisfies $g_i = \operatorname{sgn}(w_i)$ if $w_i \neq 0$, and $g_i \in [-1, 1]$ otherwise. A vector $\boldsymbol{w}(\lambda)$ is optimal if and only if it satisfies the condition:

$$0 \in \nabla f(\boldsymbol{w}(\lambda)) + \lambda \partial \|\boldsymbol{w}(\lambda)\|_1, \quad \text{i.e.,} \quad \|\nabla f(\boldsymbol{w}(\lambda))\|_\infty \leq \lambda. \tag{2}$$

Proximal Gradient Descent (PGD) solves the optimization problem in Equation (1) iteratively via:

$$\boldsymbol{w}_{t+1} = \operatorname{prox}_{\eta_t \lambda} \left( \boldsymbol{w}_t - \eta_t \nabla f(\boldsymbol{w}_t) \right), \tag{3}$$

with a step size $\eta_t = \eta < \frac{2}{L}$ to ensure convergence, where the proximal operator associated with the $\ell_1$-norm is the soft-thresholding function, given component-wise by $[\operatorname{prox}_\alpha(\boldsymbol{z})]_i = \operatorname{sgn}(z_i) \max \{ |z_i| - \alpha, 0 \}, \ \forall i \in [n]$.

The convergence properties of PGD depend critically on the assumptions regarding $f$. When $f$ is $\mu$-strongly convex (for $\mu > 0$), PGD achieves a linear (geometric) convergence rate (Beck & Teboulle, 2009; Combettes & Pesquet, 2011) with a contraction factor (if $\eta_t = \frac{1}{L}$) of $\rho = \sqrt{1 - \mu/L}$, i.e.,

$$\|\boldsymbol{w}_{t+1} - \boldsymbol{w}(\lambda)\|_2 \leq \rho \|\boldsymbol{w}_t - \boldsymbol{w}(\lambda)\|_2. \tag{4}$$

If $f$ is convex but not strongly convex (i.e., $\mu = 0$), PGD converges at a rate of $O(1/t)$ for the objective value. Accelerated methods, such as the Fast Iterative Soft-Thresholding (FISTA) (Beck & Teboulle, 2009), achieve an improved $O(1/t^2)$ convergence rate for the objective value:

$$F(\boldsymbol{w}_t; \lambda) - F(\boldsymbol{w}(\lambda); \lambda) \leq \frac{2L \|\boldsymbol{w}_0 - \boldsymbol{w}(\lambda)\|_2^2}{(t+1)^2}.$$

In many applications, only some stochastic estimates $\nabla f(\boldsymbol{w}_t, \boldsymbol{\xi}_t)$ of the gradient $\nabla f(\boldsymbol{w}_t)$ are accessible, where $\boldsymbol{\xi}_t$ is a random variable. In the strongly convex setting, under standard assumptions on the stochastic gradients (e.g., unbiasedness, and bounded variance), Stochastic Proximal Gradient

Descent (SPGD) achieves a sublinear convergence rate in expectation. This rate is typically (Rosasco et al., 2014):

$$\mathbb{E}\left[\|\boldsymbol{w}_t - \boldsymbol{w}(\lambda)\|_2^2\right] \leq \frac{C}{\mu t},$$

for some constant $C$ that depends on problem parameters. These standard results are derived and discussed extensively in foundational texts and surveys by Beck & Teboulle (2009); Nesterov (2014); Rosasco et al. (2014); Bottou et al. (2018).

## 3 METHOD

Building on the surrogate objective in Equation (1), our goal is to find a minimizer $\boldsymbol{w}_\kappa$ of $f$ subject to the sparsity constraint $\|\boldsymbol{w}\|_0 \leq \kappa$, possibly under an additional structured sparsity constraint. The surrogate goal is to choose a regularization weight $\lambda_\kappa$ that induces the (unstructured) cardinality constraint $\|\boldsymbol{w}(\lambda_\kappa)\|_0 \leq \kappa$ (see Section 3.4 for structured patterns).

Throughout, bold symbols denote vectors and vector-valued functions, while regular symbols denote scalars and scalar-valued functions ($w_i$ is component $i$ of $\boldsymbol{w}$). We denote the support of $\boldsymbol{w} \in \mathbb{R}^n$ by $\operatorname{supp}(\boldsymbol{w}) \coloneqq \{i \in [n] : w_i \neq 0\}$ and its complement by $\mathcal{Z}(\boldsymbol{w}) \coloneqq [n] \setminus \operatorname{supp}(\boldsymbol{w})$.

Denote the solution map $\boldsymbol{w}(\lambda) \coloneqq \arg\min_{\boldsymbol{w}} F(\boldsymbol{w}; \lambda)$. We apply PGD under the following assumption, which ensures that $\boldsymbol{w}(\lambda)$ is single-valued.

**Assumption 1.** *The function $f : \mathbb{R}^n \to \mathbb{R}$ is $\mu$-strongly convex and $L$-smooth with $L, \mu > 0$.*

The strong convexity in Assumption 1 yields two properties crucial to our framework (proofs in Appendix B.2):

**Lemma 1** (Compactness of the solution path). *Let $\mathcal{W}_\Lambda = \{\boldsymbol{w}(\lambda) \mid \lambda \geq 0\}$ denote the solution path. Under Assumption 1, $\mathcal{W}_\Lambda$ is compact: it is a closed set, and $\exists R > 0$ such that $\|\boldsymbol{w}(\lambda)\|_2 \leq R$ for all $\lambda \geq 0$.*

**Lemma 2** (Lipschitz continuity in $\lambda$). *The solution map $\boldsymbol{w} : \mathbb{R}_+ \to \mathbb{R}^n$ is $L_{\boldsymbol{w}}$-Lipschitz with respect to $\lambda$ in the Euclidean norm, with $L_{\boldsymbol{w}} = \frac{\sqrt{n}(L+\mu)}{\mu L}$.*

The domain of $f$ can be any closed and convex set $\mathcal{W}$ with $\boldsymbol{0}_n \in \mathcal{W}$, under the assumption that $\mathcal{W}_\Lambda$ is included in its interior $\operatorname{int}(\mathcal{W})$, so that we can get the simplified first-order Karush-Kuhn-Tucker (KKT) optimality conditions for $F(\cdot; \lambda)$ as follows:

$$|\nabla_i f(\boldsymbol{w}(\lambda))| = \lambda, \quad i \in \operatorname{supp}(\boldsymbol{w}(\lambda)); \quad |\nabla_i f(\boldsymbol{w}(\lambda))| \leq \lambda, \quad i \in \mathcal{Z}(\boldsymbol{w}(\lambda)), \tag{5}$$

where $\nabla_i f(\boldsymbol{w})$ denotes the $i$-th coordinate of $\nabla f(\boldsymbol{w})$.

The inequality over $\mathcal{Z}(\boldsymbol{w}(\lambda))$ need not be strict, as equality may occur. Define $\boldsymbol{w}_{-i} \coloneqq (w_1, \ldots, w_{i-1}, 0, w_{i+1}, \ldots, w_n)$. We prove the following in Appendix B.2:

**Lemma** (Sparsity characterization). *Under Assumption 1, the following necessary and sufficient sparsity gauge holds:*

$$w_i(\lambda) = 0 \iff \left|\nabla_i f\left(\boldsymbol{w}_{-i}(\lambda)\right)\right| \leq \lambda. \tag{6}$$

While strong convexity might seem restrictive, it can be enforced without altering the support selected at optimum for a fixed $\lambda$. Consider the Tikhonov-perturbed objective $h(\boldsymbol{w}) = f(\boldsymbol{w}) + \frac{\gamma}{2}\|\boldsymbol{w}\|_2^2$ with $\gamma > 0$. Then $\nabla_i h(\boldsymbol{w}_{-i}(\lambda)) = \nabla_i f(\boldsymbol{w}_{-i}(\lambda))$, so the gauge equation 6 is unchanged. Consequently, for any $\gamma > 0$, the solution $\boldsymbol{w}^h(\lambda, \gamma) \coloneqq \arg\min_{\boldsymbol{w}} \left\{F(\boldsymbol{w}; \lambda) + \frac{\gamma}{2}\|\boldsymbol{w}\|_2^2\right\}$ has the same inactive set determined by equation 6. In Appendix C.1 we show that $\boldsymbol{w}^h(\lambda, \gamma) \to \arg\min_{\boldsymbol{w} \in \boldsymbol{w}(\lambda)} \|\boldsymbol{w}\|_2$ as $\gamma \to 0^+$, akin to the Elastic Net (Zou & Hastie, 2005), which approximates the minimum-norm element of $\boldsymbol{w}(\lambda)$.

### 3.1 STABLE REGULARIZATIONS

Once $\boldsymbol{w}(\lambda)$ is computed, $\lambda$ is adjusted so that $|\nabla_i f(\boldsymbol{w}_{-i}(\lambda))| \leq \lambda$ is satisfied by at most $\kappa$ components. However, perturbing $\lambda$ causes the coordinates of $\boldsymbol{w}(\lambda)$ to change, creating a circular dependency. Theorem 1 (proof in Appendix B.3) shows that $\lambda$ can be continuously adjusted to cross the

curve of any non-negative continuous function, establishing the existence of what we term *stable regularizations*.

**Theorem 1** (Stable regularizations). *Let $\phi : \mathbb{R}^n \to \mathbb{R}_+$ be a continuous nonnegative function. Then there exists $\lambda \geq 0$ such that $\lambda = \phi \circ \boldsymbol{w}(\lambda)$. Define the compact set of $\phi$-stable regularizations w.r.t to $f$ as:*

$$\Lambda(\phi) := \big\{ \lambda \geq 0 \,\big|\, \phi \circ \boldsymbol{w}(\lambda) = \lambda \big\}. \tag{7}$$

It should be noted that the set of $\phi$-stable regularizations is not necessarily a singleton, and even for simple quadratic $f$, the Lasso path can be non-monotone.

Let $a_{[j]}$ denote the $j$-th largest element of the vector $\boldsymbol{a} \in \mathbb{R}^n$, and define $\phi_\kappa$ as:

$$\phi_\kappa(\boldsymbol{w}) := \big[|\nabla_1 f(\boldsymbol{w}_{-1})|, |\nabla_2 f(\boldsymbol{w}_{-2})|, \ldots, |\nabla_n f(\boldsymbol{w}_{-n})|\big]_{[\kappa+1]}. \tag{8}$$

The map $\phi_\kappa$ is nonnegative and is continuous since both $\boldsymbol{w} \to \nabla_i f(\boldsymbol{w}_{-i})$ and $\boldsymbol{a} \mapsto |\boldsymbol{a}|_{[\kappa]}$ are continuous. Theorem 1 guarantees the existence of a stable regularization $\lambda \in \Lambda(\phi_\kappa)$ for which $\phi_\kappa(\boldsymbol{w}(\lambda)) = \lambda$. Applying Equation (6), we get:

$$\begin{aligned} w_i(\lambda) = 0 &\iff \big|\nabla_i f\big(\boldsymbol{w}_{-i}(\lambda)\big)\big| \leq \lambda \\ &\iff \big|\nabla_i f\big(\boldsymbol{w}_{-i}(\lambda)\big)\big| \leq \phi_\kappa(\boldsymbol{w}(\lambda)) \end{aligned} \tag{9}$$

By definition, $\phi_\kappa(\boldsymbol{w}(\lambda))$ is the $\kappa$-th largest entry among $(|\nabla_i f(\boldsymbol{w}_{-i}(\lambda))|)_{1 \leq i \leq n}$ and the inequality (9) is satisfied by at least $n - \kappa$ components, i.e., at most $\kappa$ components are active. Thus, the task of identifying $\lambda_\kappa$ can be reduced to finding a root of the scalar map $\lambda \mapsto \phi_\kappa\big(\boldsymbol{w}(\lambda)\big) - \lambda$.

In a black-box setting, computing $\phi_\kappa$ requires $|\operatorname{supp}(\boldsymbol{w}(\lambda))|$ gradient evaluations, which is costly when the support is large. We therefore introduce a cheaper surrogate $\psi_\kappa$ (proof in Appendix B.4):

**Theorem 2** (Practical characterization). *Let $\boldsymbol{w} = \boldsymbol{w}(\lambda)$. For any $i \in [n]$, if $\exists \alpha > 0$ such that $|\nabla_i f(\boldsymbol{w}) - \alpha w_i| \leq \lambda$ then $w_i = 0$. Consequently, for any $\boldsymbol{\alpha} \in \mathbb{R}_{>0}^n$, the function $\psi_{\kappa,\boldsymbol{\alpha}} : \boldsymbol{w} \mapsto |\nabla f(\boldsymbol{w}) - \boldsymbol{\alpha} \odot \boldsymbol{w}|_{[k+1]}$, where $\odot$ is the element-wise product, satisfies:*

$$\lambda \in \Lambda(\psi_{\kappa,\boldsymbol{\alpha}}) \implies \|\boldsymbol{w}(\lambda)\|_0 \leq \kappa. \tag{}$$

In the remainder of the paper, the hyperparameter $\boldsymbol{\alpha} \in \mathbb{R}_{>0}^n$ is fixed and dropped from notations. Note that the sparsity characterization from Theorem 2 does not require the strict convexity of $f$, and $\psi_\kappa$ has the nice property of being Lipschitz continuous:

**Lemma 3** (Lipschitzness of $\psi_\kappa$). *$\psi_\kappa$ is $L_\psi$-Lipschitz in the Euclidean norm with $L_\psi := L + \max_i \alpha_i$.*

As shown in the proof in Appendix B.2, for all $\lambda \geq \lambda_{\max} := \|\nabla f(\boldsymbol{0}_n)\|_\infty$, we have $\boldsymbol{w}(\lambda) = \boldsymbol{0}_n$. Therefore, to find a stable regularization for $\phi_\kappa$ or $\psi_\kappa$, we could apply a bisection method initialized with the interval $[0, \lambda_{\max}]$: at each iteration, one solves the subproblem at the midpoint and subsequently shrinks the interval. This method yields an $\epsilon$-accurate root with a total complexity of $O(\log^2(1/\epsilon))$, requiring $O(\log(1/\epsilon))$ linear-rate PGD subproblems. In the next section, we introduce a flexible scheme that avoids fully computing $\boldsymbol{w}(\lambda)$ for each trial of $\lambda$ and operates effectively in the stochastic setting.

## 3.2 Adaptive Soft-Thresholding Algorithm (ASTRA)

Given the integer $\kappa > 0$ for the targeted sparsity level (which is then omitted from notations), set:

$$\Psi(\lambda) := \psi\big(\boldsymbol{w}(\lambda)\big), \quad \text{and} \quad \Phi(\lambda) := \Psi(\lambda) - \lambda.$$

We propose an Adaptive Soft-Thresholding Algorithm (ASTRA) for exact gradients:

$$\lambda_{t+1} = \Pi_{[0,\lambda_{\max}]} \left[ (1 - \beta_t)\lambda_t + \beta_t \psi(\boldsymbol{w}_t) \right], \tag{10}$$

$$\boldsymbol{w}_{t+1} = \operatorname{prox}_{\eta\lambda_{t+1}}\big(\boldsymbol{w}_t - \eta\nabla f(\boldsymbol{w}_t)\big), \tag{11}$$

where $\Pi_{\mathcal{C}}$ is the projection operator onto a set $\mathcal{C}$, and $\beta_t$ is a suitably chosen parameter.

Using the tracking errors $\delta_t := \|\boldsymbol{w}_t - \boldsymbol{w}(\lambda_t)\|$ and $\epsilon_t := \psi(\boldsymbol{w}_t) - \Psi(\lambda_t)$, the $\lambda$-update becomes:

$$\lambda_{t+1} = \Pi_{[0,\lambda_{\max}]} \left[ \lambda_t + \beta_t(\Phi(\lambda_t) + \epsilon_t) \right]. \tag{12}$$

This is similar to a Robbins-Monro scheme with deterministic perturbation $\epsilon_t$. For any convergence guarantees of the proposed algorithm, the iterates are required to be bounded, which is proven in Appendix B.5 in two steps:

**Lemma 4** (Decay Rate of $\delta_t$). *Let $\rho = \sqrt{1 - \mu/L}$ and $\eta \in (0, 1/L)$. The error $\delta_t$ satisfies:*

$$\delta_{t+1} \leq \rho(1 + \beta_t L_{\boldsymbol{w}} L_\psi)\delta_t + \rho L_{\boldsymbol{w}} \beta_t |\Phi(\lambda_t)| \tag{13}$$

**Corollary 1** (Boundedness of iterates). *Let $\overline{\beta} = \sup_t \beta_t$ such that $\rho(1 + \overline{\beta} L_{\boldsymbol{w}} L_\psi) < 1$, then the sequence $(\boldsymbol{w}_t)_t$ generated by the iteration from Equation (11) is bounded, and $\delta_t = O(\beta_t)$.*

Our strategy of using $\ell_1$ penalty to induce sparsity comes at the price of introducing a bias in the solution. By targeting the minimal stable regularization, $\lambda_\star$, this bias is minimized:

$$\lambda_\star := \min_{\lambda \in \Lambda(\psi)} \lambda = \min \left\{ \lambda \in \mathbb{R}_+ \mid \Phi(\lambda) = 0 \right\}. \tag{14}$$

We will adopt the following stability assumption around $\lambda_\star$:

**Assumption 2** (Asymptotic stability $\lambda_\star$). *We assume that $\lambda_\star$ is asymptotically stable for the ordinary differential equation (ODE) $\dot{\lambda} = \Phi(\lambda)$: there exist a constant $c > 0$ and a neighborhood $\mathcal{N}$ of $\lambda_\star$ such that for all $\lambda \in \mathcal{N}$:*

$$(\lambda - \lambda_\star)(\Phi(\lambda) - \Phi(\lambda_\star)) \leq -c(\lambda - \lambda_\star)^2 \tag{15}$$

Assumption 2 ensures that the equilibrium $\lambda_\star$ of the ODE $\dot{\lambda} = \Phi(\lambda)$ is locally attractive around $\lambda_\star$. This condition implies that the curve of $\Phi$ must properly cross the $\lambda$-axis at $\lambda_\star$ with a negative slope, locally, and that $|\Phi(\lambda)|$ is bounded below by $c|\lambda - \lambda_\star|$ in a neighborhood of $\lambda_\star$. If, for instance, $\Phi(\lambda)$ only "touched" zero at $\lambda_\star$ from one side ($\Phi(\lambda_\star) = 0$ but $\Phi(\lambda)$ does not change sign), or if $\Phi(\lambda)$ crossed zero but too "flatly" (e.g., if $\Phi'(\lambda_\star) = 0$ in a differentiable case), then the stability and convergence behavior around $\lambda_\star$ would be dictated by higher-order, non-linear terms, which would result in considerably slower rates. Our assumption, a standard for achieving $O(1/t)$ rates (Polyak & Juditsky, 1992; Borkar, 2008), guarantees robust and sufficiently fast local convergence.

The sequence $(\beta_t)_t$ is chosen to satisfy the Robbins-Monro conditions: $\sum_{t=0}^\infty \beta_t = \infty$ and $\sum_{t=0}^\infty \beta_t^2 < \infty$, to constitute a non-uniform infinite timeline and ensure that the perturbation of the ODE is controllable, i.e., $\sum_t \beta_t \delta_t^2 < \infty$, while being small enough to not skip the stable neighborhood of $\lambda_\star$. Since we will use the initialization $\lambda_0 = 0$, the attractive neighborhood is reached in finite time, and with a suitable choice of $(\beta_t)$, we can guarantee that the iterates remain in such neighborhood.

**Theorem 3** (Sublinear Convergence Rate). *Let $\beta_t = \beta_0/(t + t_0)$ for $t \geq 0$, with $\beta_0 > 0$ and $t_0 > 0$. Assume $\lambda_\star$ satisfies Assumption 2. Then, for a sufficiently large $\beta_0$, and $t_0$ chosen large enough so that $\beta_t < \min\left\{1, \frac{\rho - 1}{\rho L_{\boldsymbol{w}} L_\psi}\right\}$, then the iterates converge to the solution with rates*

$$\|\lambda_t - \lambda_\star\|^2 = O(1/t), \quad \|\boldsymbol{w}_t - \boldsymbol{w}(\lambda_\star)\|^2 = O(1/t). \tag{16}$$

See Appendix B.6 for the proof.

ASTRA dynamics are mainly controlled by the convergence of the $\lambda$-update in $O(1/t)$, even if the $\boldsymbol{w}$-update can be exponentially faster. We show in Appendix C.2 that if $\Psi$ is contractive around $\lambda_\star$, the following intermittent schedule converges linearly to a neighborhood of $\lambda_\star$ of radius $O(\rho^T)$:

- $\lambda_{k+1} = (1 - \beta)\lambda_k + \beta\psi(\boldsymbol{w}_{kT})$, using constant step size $\beta$.
- For $0 \leq j < T - 1$: $\boldsymbol{w}_{kT+j+1} = \text{prox}_{\eta\lambda_{k+1}}(\boldsymbol{w}_{kT+j} - \eta\nabla f(\boldsymbol{w}_{kT+j}))$.

It should be noted that to properly implement the intermittent schedule, we also need to know some properties of $f$ (i.e., $\mu$ and $L$), which is not always accessible, especially in a black-box setting. Furthermore, the necessary conditions for $\Psi$ to be contractive are not clear; however a sufficient condition is the separability of $f$, i.e., $f(\boldsymbol{w}) = \sum_{i=1}^n f_i(w_i)$.

### 3.3 STOCHASTIC ADAPTIVE SOFT-THRESHOLDING ALGORITHM (SASTRA)

We now consider the case where only stochastic estimates of $\nabla f(\boldsymbol{w}_t)$ are available. Following the formulation from Equation (12), the stochastic extension is seen as two-timescale stochastic approximation of the ODE $(\dot{\lambda}, \dot{\boldsymbol{w}}) = (\Phi(\lambda), G(w))$ where $G_t(\boldsymbol{w}) = \boldsymbol{w}_t - \text{prox}(\boldsymbol{w}_t - \eta_t f(\boldsymbol{w}))$.

The main challenge facing a stochastic ASTRA (SASTRA) is the nonlinearity of the order statistics, which introduces as bias in the estimation of our sparsity gauge $\psi(\boldsymbol{w}_t)$. We will assume that we have access to an unbiased mean map $\bar{\psi}_t$, which can be approximated practically via exponential moving average of the gradients. Let $(\mathcal{F}_t)$ be the natural filtration and assume the following:

**Assumption 3** (Unbiasedness). *The stochastic gradient $g_t$ is unbiased with bounded variance:* $\mathbb{E}\left[g_t \mid \mathcal{F}_t\right] = \nabla f(x_t)$ *and* $\mathbb{E}\left[\|g_t - \nabla f(x_t)\|^2 \mid \mathcal{F}_t\right] \leq \sigma^2$.

**Assumption 4** (Bounded variance). *There is a mean map $\bar{\psi}_t$ and noise $\xi_t$ such that $\bar{\psi}_t = \psi(\boldsymbol{w}_t) + \xi_t$, $\mathbb{E}\xi_t \mid \mathcal{F}_t = 0$, $\mathbb{E}\left[\xi_t^2 \mid \mathcal{F}_t\right] \leq \sigma_\psi^2$.*

Consider the two-time-scale iteration

$$\lambda_{t+1} = \Pi_{[0,\lambda_{\max}]}\left[(1-\beta_t)\lambda_t + \beta_t\,\psi_t\right], \qquad \boldsymbol{w}_{t+1} = \operatorname{prox}_{\eta_t\lambda_{t+1}}(\boldsymbol{w}_t - \eta_t g_t), \qquad (17)$$

The Proximal Stochastic Gradient Descent (ProxSGD) step ($\boldsymbol{w}$-update) targets the slowly moving $\boldsymbol{w}(\lambda_t)$, for which the tracking error $E[\|\boldsymbol{w}_t - \boldsymbol{w}(\lambda_t)\|^2]$ is typically $O(1/t)$.

**Theorem 4** (Convergence with log-slowed $\beta_t$). *Let $\eta_t = \frac{\eta_0}{(t+t_0)}$ and:*

$$\beta_t = \frac{\beta_0}{(t + t_0)\,(\log(t + t_0))^q}, \qquad q > \frac{1}{2},$$

*then $\sum_t \beta_t^2/\eta_t < \infty$ and $\sum_t \eta_t^2 < \infty$. Then $(\boldsymbol{w}_t, \lambda_t)$ converges a.s with mean rates:*

$$\mathbb{E}\left[\|\boldsymbol{w}_t - \boldsymbol{w}(\lambda_t)\|^2\right] = O(1/t), \quad \mathbb{E}\left[|\lambda_t - \lambda_\star|^2\right] = O(\beta_t).$$

We break down the result in Appendix B.7, which is largely inspired by Theorem 3 and relies on the Robbins-SigmundRobbins & Siegmund (1971) lemma, which requires $\sum_t \beta_t^2/\eta_t < \infty$. To fulfill such a condition, we opt for logarithmic slowed $\beta_t$. The other option is to simply choose $\eta_t = \Theta(t^a)$ for $a \in (\frac{1}{2}, 1)$, but this implies slower convergence of $(\boldsymbol{w}_t)_t$.

### 3.4 STRUCTURED SPARSE TRAINING

Although unstructured sparsity can achieve high *theoretical* compression rates, its irregular nature offers limited practical speedups on modern hardware (Jaiswal et al., 2023). Structured sparsity overcomes this by removing entire groups of parameters (Xie et al., 2023), which makes the resulting models amenable to hardware acceleration and efficient memory access. ASTRA naturally extends to structured sparsity by applying its adaptive mechanism to groups of parameters rather than individual weights. This is achieved by employing a group $\ell_1$ regularization penalty (Mairal et al., 2011) to encourage sparsity at the group level (Bach et al., 2012), using $\ell_2$-norm within a group.

In order to represent the different sparsity patterns, we introduce a Structured Sparsity Algebra (SSA) in Appendix C.3. Here, we formulate the regularization structure that is adapted to SSA.

Let $\mathcal{B} = \{\boldsymbol{b}_1, \ldots, \boldsymbol{b}_{pq}\}$ be a partition of $[n]$ into disjoint $pq$ blocks. We then define the group grid $\mathcal{G} = \{\boldsymbol{g}_1, \ldots, \boldsymbol{g}_q\}$, consisting of $q$ disjoint groups $\boldsymbol{g}_i = \{\boldsymbol{b}_{(i-1)q+1}, \ldots, \boldsymbol{b}_{iq+p}\}$. To extend our framework to induce $\kappa$-sparse groups, we replace our unstructured $\ell_1$ regularization with the gauge function

$$\Omega_{\mathcal{B},\mathcal{G}}(\boldsymbol{w}) \coloneqq \sum_{i=1}^{q} \lambda_i \sum_{\boldsymbol{b} \in \boldsymbol{g}_q} \|\boldsymbol{w}_{\boldsymbol{b}}\|,$$

then $\lambda_i$ is dynamically adjusted using ASTRA to keep at most $\kappa$ blocks per group. The regularization is now symmetric across groups. We consider the case with a single group ($q = 1$).

The proximal operator for such block-wise regularization is

$$\operatorname{prox}_\alpha(\boldsymbol{w}_{\boldsymbol{b}_j}) = \max\left\{0, 1 - \frac{\alpha}{\|\boldsymbol{w}_{\boldsymbol{b}_j}\|_2}\right\} \boldsymbol{w}_{\boldsymbol{b}_j}.$$

The optimality conditions are then extended as $\nabla_{\boldsymbol{b}_j} f(\boldsymbol{w}^*) \in \lambda \partial \|\boldsymbol{w}_{\boldsymbol{b}_j}^*\|_2$, with $\partial \|\boldsymbol{w}_{\boldsymbol{b}}\|_2 = \frac{\boldsymbol{w}_{\boldsymbol{b}}}{\|\boldsymbol{w}_{\boldsymbol{b}}\|_2}$ for an active group, and $\partial \|\boldsymbol{w}_{\boldsymbol{b}}\|_2 \in B_{|\boldsymbol{b}|}(0, 1)$ for inactive ones.

The stable regularizations extend smoothly; the characterization from Equation (6) becomes:

$$\|\boldsymbol{w_b}(\lambda)\| = 0 \iff \|\nabla_{\boldsymbol{b}} f(\boldsymbol{w}_{-\boldsymbol{b}}(\lambda))\| \leq \lambda. \tag{18}$$

We write $S_{++}^{\mathcal{B}}$ for the set of block-diagonal SPD matrices conformable with the block partitioning:

$$\mathcal{S}_{++}^{\mathcal{B}} := \left\{ \boldsymbol{A} = \text{blkdiag}(\boldsymbol{A}_1, \ldots, \boldsymbol{A}_m) \mid \boldsymbol{A}_j \in \mathbb{R}^{|\boldsymbol{b}_j| \times |\boldsymbol{b}_j|}, \boldsymbol{A}_j \succ 0 \right\}$$

**Theorem 5** (Practical characterization 2). *If $\exists \boldsymbol{A} \in \mathcal{S}_{++}^{|\boldsymbol{b}|}$ such that $\|\nabla_{\boldsymbol{b}} f(\boldsymbol{w}(\lambda)) - \boldsymbol{A}\boldsymbol{w_b}(\lambda)\| \leq \lambda$ then $\|\boldsymbol{w_b}\| = 0$. Hence, it follows that for $\psi_\kappa^{(\boldsymbol{A})} : \boldsymbol{w} \mapsto [\boldsymbol{u}(\boldsymbol{w})]_{[\kappa+1]}$ with $\boldsymbol{u}_j(\boldsymbol{w}) = \|\nabla_{\boldsymbol{b}_j} f(\boldsymbol{w}) - \boldsymbol{A}_j\boldsymbol{w}_{\boldsymbol{b}_j}\|$ and $\boldsymbol{A} \in \mathcal{S}_{++}^{\mathcal{B}}$, if $\lambda \in \Lambda(\psi_\kappa^{(\boldsymbol{A})})$ then the solution $\boldsymbol{w}(\lambda)$ has at most $\kappa$ active blocks, i.e. $\text{nnz}\big(\{\|\boldsymbol{w}_{\boldsymbol{b}_1}(\lambda)\|, \ldots, \|\boldsymbol{w}_{\boldsymbol{b}_p}(\lambda)\|\}\big) \leq \kappa$.*

While Theorem 5 is more general than the unstructured characterization of Theorem 2, we can still use $\boldsymbol{A} = \text{diag}(\boldsymbol{\alpha}) \in \mathcal{S}_{++}^{\mathcal{B}}$ for $\boldsymbol{\alpha} \in \mathbb{R}_{>0}^n$ while keeping the same guarantees (proof in Appendix B.8).

# 4 RELATED WORK

A first-order Taylor expansion of the optimality condition in Equation (6) around $\boldsymbol{w}$ gives rise to a set of saliency scores. Specifically, using $H$, the Hessian of $f$, the saliency of a weight $w_i$ is is evaluated as:

$$s_i := |\nabla_i f(\boldsymbol{w}_{-i})| \approx |\nabla_i f(\boldsymbol{w}) - H_{i,i}(\boldsymbol{w})w_i|, \tag{19}$$

where

**Connections to Foundational Pruning Methods.** Our framework recovers and generalizes several classic pruning criteria. For example, the OBD (LeCun et al., 1989) criterion, which estimates the change in the objective function $\delta F_i$ from pruning a single weight $w_i$, is given by:

$$\delta F_i \approx -\left( \nabla_i f(\boldsymbol{w})w_i - \frac{1}{2}H_{i,i}w_i^2 \right). \tag{20}$$

The derivative of this saliency with respect to $w_i$ is precisely our score, that is, $s_i = |\frac{d}{dw_i}\delta F_i|$. Crucially, our derivation of $s_i$ does not require the strong assumptions of OBD, namely that the model is fully converged ($\nabla f(\boldsymbol{w}) = \boldsymbol{0}$) or that the Hessian is diagonal. Similarly, OBS (Hassibi et al., 1993), which analytically computes the change in objective after optimally re-adjusting all other weights, can be seen as a more complex prune-finetune heuristic. While OBS relies on inverting the Hessian to find the optimal compensation, ASTRA achieves a similar result iteratively during training based on gradients.

**Unifying Modern Pruning Heuristics.** ASTRA provides a theoretical grounding for modern empirically driven methods. For example, Rigging The Lottery (RigL) (Evci et al., 2020) implements a dynamic train-prune-grow cycle where weights with the lowest magnitude are pruned and inactive weights with the largest gradient are grown. This heuristic can be interpreted as an instance of AS-TRA: when the weight multiplier $\alpha_i$ in our framework is large such that $|\alpha_i w_i| \gg |\nabla_i f(\boldsymbol{w})|$, the saliency scores for active weights are dominated by their magnitude. For inactive weights ($w_i = 0$), the scores reduce to the gradient magnitudes $(|\nabla_i f(\boldsymbol{w})|)_i$, mirroring RigL's grow phase.

Furthermore, our framework provides a theoretical grounding for recent one-shot pruning methods for LLMs. For the standard layer-wise reconstruction objective, $f(\boldsymbol{W}) = \|\boldsymbol{X}\boldsymbol{W} - \boldsymbol{Y}\|^2$, used by methods like Wanda (Sun et al., 2024) and SparseGPT (Frantar & Alistarh, 2023), the Hessian diagonals are $H_{j,j} = \|\boldsymbol{X}_j\|_2^2$. Under typical initialization conditions where $\nabla f(\boldsymbol{W}_0) = \boldsymbol{0}$, our general saliency score from Equation (19) simplifies to $s_{ij} = |W_{ij}|\|\boldsymbol{X}_j\|_2^2$. This is a direct variant of the empirical Wanda heuristic, which uses $|W_{ij}|\|\boldsymbol{X}_j\|_2$. SparseGPT, in contrast, uses a more complex iterative process based on OBS. It greedily removes weights and then analytically updates the remaining ones to compensate for the change in the objective based on the inverse of the Hessian, while ASTRA uses gradient descent to continuously adapt weights throughout the training process.

**Algorithm 1** Stochastic ASTRA Algorithm

**Require:** Loss function $\mathcal{L}$, optimizer $opt$, parameter $\boldsymbol{w} \in \mathbb{R}_n$, sparsity level $\kappa$, $\boldsymbol{\alpha} \in \mathbb{R}_{>0}^n$, grids $\mathcal{B}, \mathcal{G}$, timescales $(\eta_t)_t$, $(\beta_t)_t$, EMA decay $\rho$.

$t \leftarrow 0$

Initialize: $\boldsymbol{m}_t \leftarrow \boldsymbol{0}_n$, $\lambda_t^{\mathcal{G}} \leftarrow \boldsymbol{0}_q$.

**while** not terminated **do**
    Sample batch $b_t$.
    Get (pseudo) gradient $g_t = opt(\mathcal{L}(b_t))$
    $\boldsymbol{m}_t \leftarrow (1-\rho)\boldsymbol{m}_{t-1} + \rho g_t$
    compute $\psi_{\mathcal{G}} = \psi_\kappa^{(\boldsymbol{\alpha})}(\boldsymbol{w}, \boldsymbol{m}_t)$ per group
    $\lambda_t \leftarrow (1-\beta_t)\lambda_{t-1} + \beta_t \psi_{\mathcal{G}}$
    $\boldsymbol{w} \leftarrow \mathrm{prox}_{\eta_t \lambda_t, \mathcal{G}}(\boldsymbol{w} - \eta_t g_t)$
    $t \leftarrow t + 1$
**end while**

**Algorithm 2** Bonsai Networks

**Require:** Neural network $\mathcal{N}_{\boldsymbol{\theta}}$ parameterized by $\boldsymbol{\theta}$, SGD optimizer $opt$, warm-up time $T_w$, freeze time $T_f$.

**for** $t = 1 \ldots T_w$ **do**
    Update $\boldsymbol{\theta}$ using SGD optimizer.
    Update SASTRA states $\boldsymbol{m}_t$ and $\lambda_t^{\mathcal{G}}$.
**end for**
**for** $t = T_w + 1 \ldots T_f$ **do**
    Update $\boldsymbol{\theta}$ using SGD optimizer.
    Update SASTRA states $\boldsymbol{m}_t$ and $\lambda_t$.
    Soft-threshold using $\lambda_t$
**end for**
Freeze $\mathrm{supp}(\boldsymbol{\theta})$ using magnitude pruning.
**for** $t = T_f + 1 \ldots T$ **do**
    Apply SGD steps to $\mathrm{supp}(\boldsymbol{\theta})$
**end for**

Figure 1: The core algorithms used to train sparse neural networks with provided sparsity Structure using Stochastic ASTRA iterations.

## 5 EXPERIMENTS

Our adaptive regularization approach is, at its core, a sparse-coding–inspired method. We therefore first benchmark it against its closest counterpart, Iterative Hard Thresholding (IHT). In particular, our ASTRA procedure of identifying a stable support and then updating the active weights without further thresholding; can be viewed as an IHT-style heuristic. We report results in Appendix A.1, comparing ASTRA to IHT and Optimal Brain Damage (OBD), which is equivalent to WANDA in the one-dimensional output case. The results show that ASTRA delivers consistently strong performance across settings, especially at high sparsity.

### 5.1 BENCHMARKING SPARSE TRAINING

We extend our approach to sparsely train a *Bonsai* ResNet-32 on CIFAR-10 and CIFAR-100 at high sparsity, following the iterations detailed in Figure 1. We use CIFAR-10/100 (50k train / 10k test, $32 \times 32$). Per-channel normalization (dataset-specific mean/std) is applied. No external data are used; augmentations are limited to random horizontal flip and random crop with 4-pixel padding. Test-time evaluation uses a single center crop (the identity at $32 \times 32$).

Our primary backbone is ResNet-32 as introduced by Wang et al. (2020). All convolutional and linear layers are candidates for pruning in the unstructured case. In the channel structured sparse training, where we sparsify at the filter level, we aditionally exclude the first convolution, the final classifier.

We consider two variants:

- **Unstructured:** block size $(1, 1, 1, 1)$ with global coupling across network weights as defined by our Structured Sparsity Algebra, excluding batch-normalization parameters. This *unstructured Bonsai ResNet-32* outperforms several established sparse-training approaches.

- **Structured:** channel-wise pruning that removes entire filters at once (excluding the input convolution and downsampling layers), trading flexibility for hardware alignment; this variant trails slightly due to the reduced degrees of freedom when selecting the sparsity mask.

We compare against various sparsity inducing techniques, including Magnitude Pruning (MP), OBD, and Lottery Ticket (LT), Gradient Signal Preservation Criterion (GraSP) (Wang et al., 2020),

Sparse Evolutionary Training(SET)(Mocanu et al., 2017), Dynamic Sparse Reparameterization (DSR)(Mostafa & Wang, 2019), and the classical Iterative Hard Thresholding (IHT),

Table 1: Test accuracy of ResNet32 on CIFAR-10 and CIFAR-100. Starred methods are reproduced results. Non-stared methods have scores as were self-reported in the literature (Wang et al., 2020).

| Paradigm | Method | CIFAR-10 | | CIFAR-100 | |
|---|---|---|---|---|---|
| | | 90% | 95% | 90% | 95% |
| Initial | SNIP | 92.59 | 91.01 | 68.89 | 68.89 |
| | GraSP⋆ | 92.38 | 91.39 | 69.24 | 66.59 |
| Pruning | OBD | 94.17 | 93.29 | 71.46 | 68.73 |
| | MP Prune | 94.21 | 93.02 | 72.34 | 67.38 |
| | LT | 93.31 | 91.06 | 68.99 | 66.12 |
| DST | DSR | 92.97 | 91.61 | 69.63 | 68.20 |
| | SET | 92.30 | 90.75 | 69.66 | 67.10 |
| | IHT⋆ | 92.29 | 91.60 | 68.64 | 67.03 |
| | Bonsai | **93.05** | **92.72** | **71.89** | **71.60** |
| | Bonsai-Channel | 91.11 | 90.23 | 69.53 | 67.49 |
| Dense | Baseline⋆ | 94.32 | | 74.44 | |

## 5.2 BENCHMARKING STRUCTURED PRUNING

For structured pruning, we test on pruning the LM head of Qwen 3 - 8B LLM (Yang et al., 2025) on wikitext data. To do so, we take 16000 tokens from wikitext data that we project into the vocabulary space. The goal then is to prune the head in a structured way to gain on memory and space. We target 75% sparsity, under the criterion of minimizing the KL divergence between the original output logits and those of our pruned layer. Our goal is to achieve a 4:16 block-sparsity with block-size $16 \times x16$ along the K dimension. In this case, the layer weight shape is $4096 \times 151936$, for a total of 622M parameters and 2.4M blocks. Our goal is to leverage the Dense$\times$ 4:16 Block-Sparse kernel in Appendix C.4, which achieve a $\times 1.64$ speed up on Nvidia A100 GPU when the input is column major. Using Wanda to prune the layer (without structure) yields a mean sample KL divergence of 0.016 versus 0.08 in unstructured block- sparse and 0.10 for our 4:16 Block-Sparse structure. In contrast, the unstructured block-sparse pattern at 75% sparsity achieves $\times 0.7$ performance compared to $\times 1.64$ using our structure w.r.t to the dense matrix multiplication.

## 6 CONCLUSION AND DISCUSSION

We introduced ASTRA, an adaptive soft–thresholding framework that drives a model toward a *target* sparsity pattern by updating the regularization strength online. By casting target sparsity selection as a scalar root–finding problem over the regularization weight and tracking it with a two time–scale recursion, we obtain principled control of sparsity with provable rates. In the deterministic setting we established sublinear (and, under an intermittent schedule, local linear) convergence of both parameters and regularization, while in the stochastic setting we proved almost–sure convergence with mean–square rates consistent with the step–size schedule. These results provide a unifying, proximal view that clarifies the assumptions behind several heuristic sparse–training methods.

Beyond theory, we demonstrated that *Bonsai* networks trained with SASTRA achieve competitive accuracy at high sparsity on CIFAR–10/100, matching or surpassing strong pruning and DST baselines, and we provided a structured LLM case study showing that hardware–aligned 4:16 block sparsity can deliver practical speedups with controlled output drift.

**Limitations and future work.** Our analysis relies on convexity, while modern deep networks are nonconvex; extending the guarantees to these relaxed conditions is a natural next step. The stochastic theory assumes an unbiased surrogate for the order–statistic gauge; quantifying robustness under bias would further strengthen the framework. Finally, while our structured experiments validate a 4:16 kernel on a single LLM head, broader system–level evaluations (end–to–end latency/throughput across kernels and hardware) and scaling experiments on larger backbones are promising directions.

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

## A NOTES ON REPRODUCIBILITY AND RESULTS

### A.1 SPARSE-CODING BENCHMARK

Since our algorithm behaves like a pseudo-IHT, we benchmark it against IHT. Our theory requires the strong convexity assumption, but in practice that is not guaranteed. We therefore adopt the accelerated FISTA for the $w$-update instead of the plain ISTA iteration, and use the accelerated point to perform the $\lambda$-update. We adopt the same benchmark as Ida et al. (2024) using gisette, ledgar, real-sim and epsilon from LIBSVM. For each density level $\kappa$, the table reports the score of method $\mathcal{M}$ as:

$$Score(\mathcal{M}) = \frac{MSE(\mathcal{M})}{MSE(\text{best method})} - 1$$

As such, the best method always has score $0$. When two methods show the same score, the one in bold is the best one, meaning that its error is within $1\%$ of the best approach.

| Dataset | Method | $\kappa$ | | | | |
|---|---|---|---|---|---|---|
| | | 1% | 10% | 25% | 50% | 75% |
| Gisette | IHT | .005 | 0.091 | .111 | .298 | 0.84 |
| | OBD | 77.50 | 25.00 | 8.16 | 2.89 | 2.73 |
| | ASTRA | **.000** | **.000** | **.000** | **.000** | **.000** |
| Epsilon | IHT | .060 | .012 | .002 | .002 | .001 |
| | OBD | .327 | .027 | .013 | **.000** | **.000** |
| | ASTRA | **.000** | **.000** | **.000** | .000 | .000 |
| Real-Sim | IHT | .084 | .162 | .161 | .150 | .050 |
| | OBD | .325 | .062 | **.000** | **.000** | **.000** |
| | ASTRA | **.000** | **.000** | .018 | .051 | .023 |
| Edgar | IHT | .085 | .052 | .068 | .066 | .055 |
| | OBD | 1.441 | .226 | .089 | **.000** | **.000** |
| | ASTRA | **.000** | **.000** | **.000** | .025 | .003 |

Table 2: Sparse Coding Benchmark: relative error w.r.t to the best performing method.

The table shows that ASTRA performs consistently better than OBD (Wanda) and IHT in high sparsity modes.

### A.2 EFFECT OF LEARNING RATE

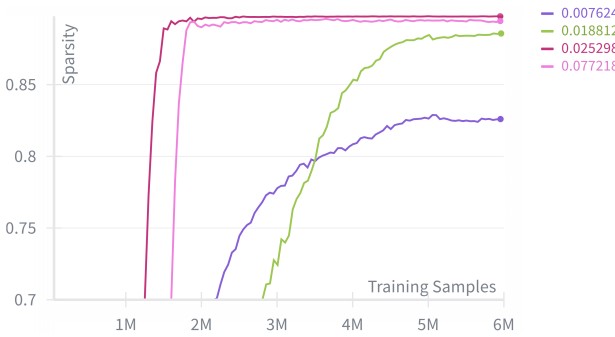

Figure 2: Reaching the target sparsity faster requires a higher learning rate, but that also results in worse performance in Neural Network training due to convergence to sub-optimal region that favors minimizing the $\ell_1$ norm over the loss function

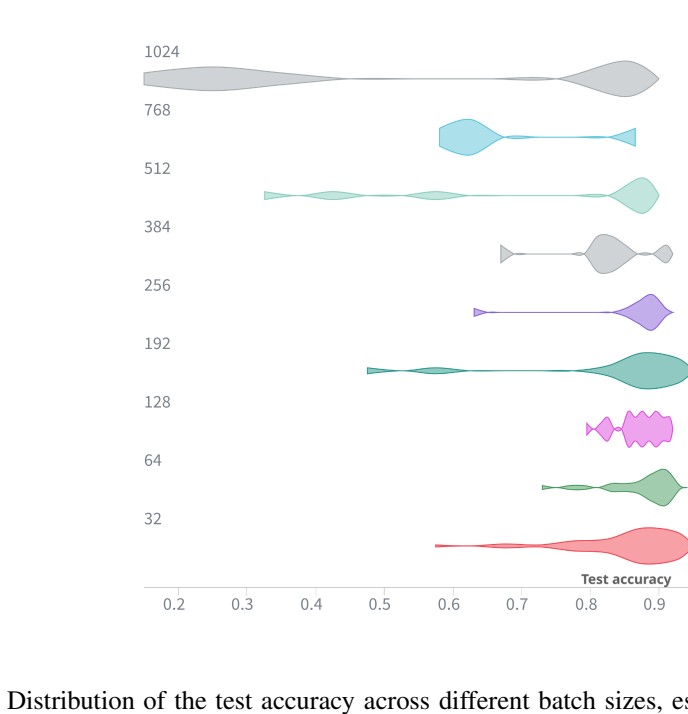

Figure 3: Distribution of the test accuracy across different batch sizes, estimated on 212 runs for arbitrary choices of hyperparameters. The batch size is the most significant factor that affects SAS-TRA performance, lower batches yield higher performance, with a -0.503 correlation with the final test accuracy.

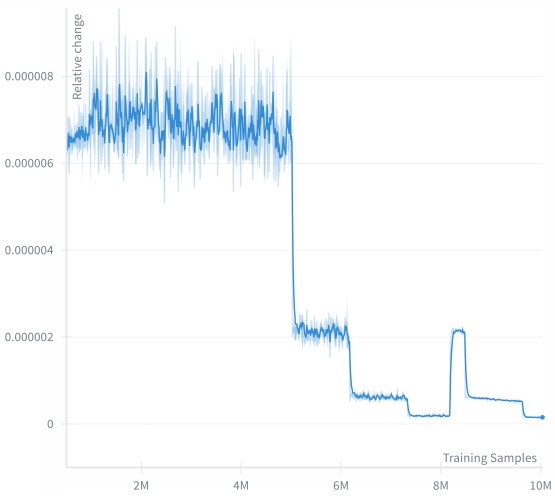

Figure 4: At training sample 8M, the support of the layer weight is frozen. In this case, The layer weight was 35% sparse before $T_f$ and 87% sparse afterwards, yet, the relative delta norm is less than 0.0003%, showing that the components of the parameter were already hovering around 0.0.

## A.3 HYPER-PARAMETERS FOR REPORTED RESULTS

These values are also provided in the configuration files of the reproducibility code. The reader can refer to the implementation for more details.

| Component | Hyper-parameter | CIFAR-10 | CIFAR-100 |
|---|---|---|---|
| Optimizer | type | SGD | SGD |
| | Initial learning rate | 0.0125 | 0.02 |
| | Momentum | 0.9437 | 0.977 |
| | Weight decay | $6.1 \times 10^{-4}$ | $2.52 \times 10^{-5}$ |
| LR Schedule | type | Multi-step | Multi-step |
| | decay LR every | 95 epochs | 47 |
| | decay rate | 0.33 | 0.33 |
| SASTRA | $\alpha_i$ | 7.66 | 9.14 |
| | $\beta$ schedule | constant | constant |
| | $\beta_t$ | 0.065 | 0.005 |
| | $\lambda_{\max}$ | 0.0015 | 0.0013 |
| | EMA $\rho$ | 0.018 | 0.004 |
| | $T_w$ | 8 | 11 |
| | $T_f$ | 150 | 132 |
| | Epochs | 200 | 150 |
| | Batch size | 192 | 32 |

Table 3: Hyper-parameters for SASTRA on CIFAR Datasets with ResNet-32

## B  PROOFS

**Notes**  For expositional clarity when analyzing the solution for a *fixed* $\lambda$, we assume, without loss of generality, that all entries in its support are positive, i.e., $w_i(\lambda) > 0$ for all $i \in \text{supp}(\boldsymbol{w}(\lambda))$. This is justified by a simple change of variables $z_i = -w_i$ for negative components $w_i(\lambda)$, which preserves the $\ell_1$-norm term ($\lambda|w_i| = \lambda|z_i|$) and the convexity of $F(\boldsymbol{w}; \lambda)$.

### B.1  PRELIMINARIES

We will make several uses of the following lemmas (see Nesterov (2014, Thm. 2.1.12) for a proof).

**Lemma B.1** (Strong monotonicity). *For a $\mu$-strongly convex function $f$ with $L$-Lipschitz continuous gradient, its gradient $\nabla f$ satisfies, for all $\boldsymbol{w}_1, \boldsymbol{w}_2$ in its domain,*

$$(\nabla f(\boldsymbol{w}_1) - \nabla f(\boldsymbol{w}_2))^\top (\boldsymbol{w}_1 - \boldsymbol{w}_2) \geq \frac{\mu L}{\mu + L} \|\boldsymbol{w}_1 - \boldsymbol{w}_2\|_2^2 + \frac{1}{\mu + L} \|\nabla f(\boldsymbol{w}_1) - \nabla f(\boldsymbol{w}_2)\|^2$$

$$\geq \frac{\mu L}{\mu + L} \|\boldsymbol{w}_1 - \boldsymbol{w}_2\|_2^2 \tag{21}$$

*In particular,*

$$\|\boldsymbol{w}_1 - \boldsymbol{w}_2\|_2 \leq \frac{1}{\mu} \|\nabla f(\boldsymbol{w}_1) - \nabla f(\boldsymbol{w}_2)\|_2 \tag{22}$$

*which holds trivially if $\boldsymbol{w}_1 = \boldsymbol{w}_2$ otherwise it follows from the Cauchy-Schwarz inequality.*

**Lemma B.2** (Zero solutions beyond). *If $\lambda \geq \|\nabla f(\boldsymbol{0}_n)\|_\infty$, then $\boldsymbol{w}(\lambda) = \boldsymbol{0}_n$.*

*Proof.* Let $\lambda \geq \|\nabla f(\boldsymbol{0}_n)\|_\infty$, then:

$$
\begin{aligned}
F(\boldsymbol{w}; \lambda) - F(\boldsymbol{0}_n; \lambda) &= (f(\boldsymbol{w}) - f(\boldsymbol{0}_n)) + \lambda\|\boldsymbol{w}\|_1 \\
&\geq \langle \nabla f(\boldsymbol{0}_n), \boldsymbol{w} \rangle + \lambda\|\boldsymbol{w}\|_1 && \text{using convexity of } f \\
&\geq -\|\nabla f(\boldsymbol{0}_n)\|_\infty \|x\|1 + \lambda\|\boldsymbol{w}\|_1 && \text{Hölder's inequality with } (p,q) = (1, \infty) \\
&= (\lambda - \|\nabla f(\boldsymbol{0}_n)\|_\infty)\|\boldsymbol{w}\|_1 \geq 0
\end{aligned}
$$

Therefore $F(\boldsymbol{w}, \lambda) \geq F(\boldsymbol{0}_n, \lambda)$ for all $\boldsymbol{w} \in \mathbb{R}^n$. If, in addition, $f$ is $\mu$-strongly convex for $\mu > 0$, then the minimizer is unique; thus $\boldsymbol{w}(\lambda) = \boldsymbol{0}_n$. $\qquad\square$

**Theorem B.1** (Robbins & Siegmund (1971, Thm. 1))**.** *Let $(\mathcal{F}_t)$ be a filtration and let $(Y_t)$, $(a_t)$, $(b_t)$, $(c_t)$ be sequences of nonnegative random variables adapted to $(\mathcal{F}_t)$ such that*

$$\mathbb{E}[Y_{t+1} \mid \mathcal{F}_t] \leq (1 + a_t)Y_t - b_t + c_t, \qquad t \geq 0,$$

*with $\sum_t a_t < \infty$ and $\sum_t c_t < \infty$ almost surely. Then $Y_t$ converges almost surely to a finite random variable $Y_\infty$, and $\sum_t b_t < \infty$ almost surely.*

### B.2 PROOF OF LEMMA 1

**Lemma 1** (Compactness of the solution path)**.** *Let $\mathcal{W}_\Lambda = \{\boldsymbol{w}(\lambda) \mid \lambda \geq 0\}$ denote the solution path. Under Assumption 1, $\mathcal{W}_\Lambda$ is compact: it is a closed set, and $\exists R > 0$ such that $\|\boldsymbol{w}(\lambda)\|_2 \leq R$ for all $\lambda \geq 0$.*

*Proof.* We start by showing boundedness. Let $\lambda_{\max} = \|\nabla f(\boldsymbol{0}_n)\|_\infty$, then by inspection of the optimality conditions (and uniqueness of the solution, guaranteed by strong convexity), we see that $\boldsymbol{w}(\lambda) = 0 \;\forall \lambda \geq \lambda_{\max}$.

Hence $\mathcal{W}_\Lambda = \{\boldsymbol{w}(\lambda) \mid 0 \leq \lambda \leq \lambda_{\max}\}$. For $f$ a strongly convex function, then $\nabla f$ is (continuously) invertible which follows from eq. (22), so $\boldsymbol{w}(\lambda) = (\nabla f)^{-1}(-\lambda s)$ for a subgradient $s$. As subgradients are bounded $\|s\|_\infty \leq 1$ and $\lambda \in [0, \lambda_{\max}]$ is bounded, and since $(\nabla f)^{-1}$ is continuous, it follows $\mathcal{W}_\lambda$ is bounded.

We now show closedness, i.e., that $\mathcal{W}_\Lambda$ contains all its limit points. Let $(\boldsymbol{w}_k)$ be a sequence in $\mathcal{W}_\Lambda$ that converges to $\overline{\boldsymbol{w}}$, and we wish to show $\overline{\boldsymbol{w}} \in \mathcal{W}_\Lambda$. If $\overline{\boldsymbol{w}} = \boldsymbol{0}_n$, then $\overline{\boldsymbol{w}} \in \mathcal{W}_\Lambda$ since $\boldsymbol{0} = \boldsymbol{w}(\lambda_{\max})$, so from now on assume $\overline{\boldsymbol{w}} \neq \boldsymbol{0}_n$. Thus for large enough $k$, $\boldsymbol{w}_k \neq \boldsymbol{0}$. Because $\boldsymbol{w}_k \in \mathcal{W}_\Lambda$, we can write $\boldsymbol{w}_k = \boldsymbol{w}(\lambda_k)$ for some $\lambda_k \geq 0$. For nonzero $\boldsymbol{w}_k$, the optimality conditions eq. (2) are equivalent to $\|\nabla f(\boldsymbol{w}_k)\|_\infty = \lambda_k$ (cf. Lemma B.2). Since $(\boldsymbol{w}_k)$ converges and since $\nabla f$ is continuous (since it is Lipschitz) and norms are continuous, it follows that the sequence $(\lambda_k)$ also converges; denote its limit by $\overline{\lambda}$. Because $\lambda_k \geq 0$ and $[0, \infty)$ is a closed set, then $\overline{\lambda} \geq 0$ as well. Furthermore, by the same continuity argument, we have $\|\nabla f(\overline{\boldsymbol{w}})\|_\infty = \overline{\lambda}$, which is the necessary and sufficient condition for optimality, so it follows $\overline{\boldsymbol{w}} = \boldsymbol{w}(\overline{\lambda})$ hence $\overline{\boldsymbol{w}} \in \mathcal{W}_\Lambda$.

$\square$

**Lemma 2** (Lipschitz continuity in $\lambda$)**.** *The solution map $\boldsymbol{w}: \mathbb{R}_+ \to \mathbb{R}^n$ is $L_{\boldsymbol{w}}$-Lipschitz with respect to $\lambda$ in the Euclidean norm, with $L_{\boldsymbol{w}} = \frac{\sqrt{n}(L + \mu)}{\mu L}$.*

*Proof.* For any subgradients $s_1 \in \partial \|\boldsymbol{w}_1\|_1$ and $s_2 \in \partial \|\boldsymbol{w}_2\|_1$, we have monotonicity:

$$(\boldsymbol{s}_1 - \boldsymbol{s}_2)^\top (\boldsymbol{w}_1 - \boldsymbol{w}_2) \geq 0 \tag{23}$$

For proofs of this inequality and a broader background on convex optimization, see Bubeck (2015).

Let $\lambda_1, \lambda_2 \geq 0$, and $\boldsymbol{w}_1 = \boldsymbol{w}(\lambda_1)$ and $\boldsymbol{w}_2 = \boldsymbol{w}(\lambda_2)$. The first-order optimality conditions imply the existence of subgradients $s_1 \in \partial \|\boldsymbol{w}_1\|_1$ and $s_2 \in \partial \|\boldsymbol{w}_2\|_1$ such that:

$$\nabla f(\boldsymbol{w}_1) = -\lambda_1 \boldsymbol{s}_1$$
$$\nabla f(\boldsymbol{w}_2) = -\lambda_2 \boldsymbol{s}_2.$$

Using the strong convexity of $f$ and Equation (21),

$$\frac{\mu L}{\mu + L} \|\boldsymbol{w}_1 - \boldsymbol{w}_2\|_2^2 \leq (\nabla f(\boldsymbol{w}_1) - \nabla f(\boldsymbol{w}_2))^\top (\boldsymbol{w}_1 - \boldsymbol{w}_2)$$

$$= (-\lambda_1 \boldsymbol{s}_1 - (-\lambda_2 \boldsymbol{s}_2))^\top (\boldsymbol{w}_1 - \boldsymbol{w}_2)$$

$$= (\lambda_2 \boldsymbol{s}_2 - \lambda_1 \boldsymbol{s}_1)^\top (\boldsymbol{w}_1 - \boldsymbol{w}_2)$$

$$= -(\lambda_1 \boldsymbol{s}_1 - \lambda_2 \boldsymbol{s}_2)^\top (\boldsymbol{w}_1 - \boldsymbol{w}_2).$$

Let's analyze the term $\lambda_1 \boldsymbol{s}_1 - \lambda_2 \boldsymbol{s}_2$:

$$\lambda_1 \boldsymbol{s}_1 - \lambda_2 \boldsymbol{s}_2 = \lambda_1 \boldsymbol{s}_1 - \lambda_1 \boldsymbol{s}_2 + \lambda_1 \boldsymbol{s}_2 - \lambda_2 \boldsymbol{s}_2 = \lambda_1 (\boldsymbol{s}_1 - \boldsymbol{s}_2) + (\lambda_1 - \lambda_2) \boldsymbol{s}_2$$

Substituting this back:

$$\frac{\mu L}{\mu + L}\|\boldsymbol{w}_1 - \boldsymbol{w}_2\|_2^2 \leq -[\lambda_1(\boldsymbol{s}_1 - \boldsymbol{s}_2) + (\lambda_1 - \lambda_2)\boldsymbol{s}_2]^\top(\boldsymbol{w}_1 - \boldsymbol{w}_2)$$
$$= -\lambda_1(\boldsymbol{s}_1 - \boldsymbol{s}_2)^\top(\boldsymbol{w}_1 - \boldsymbol{w}_2) - (\lambda_1 - \lambda_2)\boldsymbol{s}_2^\top(\boldsymbol{w}_1 - \boldsymbol{w}_2).$$

Using Equation 23, we have $(\boldsymbol{s}_1 - \boldsymbol{s}_2)^\top(\boldsymbol{w}_1 - \boldsymbol{w}_2) \geq 0$. As $\lambda_1 \geq 0$, the first term $-\lambda_1(\boldsymbol{s}_1 - \boldsymbol{s}_2)^\top(\boldsymbol{w}_1 - \boldsymbol{w}_2)$ is non-positive. Therefore:

$$\frac{\mu L}{\mu + L}\|\boldsymbol{w}_1 - \boldsymbol{w}_2\|_2^2 \leq -(\lambda_1 - \lambda_2)\boldsymbol{s}_2^\top(\boldsymbol{w}_1 - \boldsymbol{w}_2) = (\lambda_1 - \lambda_2)\boldsymbol{s}_2^\top(\boldsymbol{w}_2 - \boldsymbol{w}_1)$$

Applying the Cauchy-Schwarz inequality:

$$\frac{\mu L}{\mu + L}\|\boldsymbol{w}_1 - \boldsymbol{w}_2\|_2^2 \leq |\lambda_1 - \lambda_2| \cdot |s_2^\top(\boldsymbol{w}_1 - \boldsymbol{w}_2)| \leq |\lambda_1 - \lambda_2| \cdot \|s_2\|_2 \|\boldsymbol{w}_1 - \boldsymbol{w}_2\|_2$$

Since $\|s_2\|_\infty \leq 1$ by the definition of $\partial\|\cdot\|_1$, it follows $\|s_2\|_2 \leq \sqrt{n}$. Substituting, if $\boldsymbol{w}_1 \neq \boldsymbol{w}_2$, we can divide by $\|\boldsymbol{w}_1 - \boldsymbol{w}_2\|_2$ (if $\boldsymbol{w}_1 = \boldsymbol{w}_2$, the inequality holds trivially):

$$\frac{\mu L}{\mu + L}\|\boldsymbol{w}_1 - \boldsymbol{w}_2\|_2^2 \leq \sqrt{n}|\lambda_1 - \lambda_2|\|\boldsymbol{w}_1 - \boldsymbol{w}_2\|_2.$$

Hence

$$\|\boldsymbol{w}_1 - \boldsymbol{w}_2\|_2 \leq \frac{\sqrt{n}(L + \mu)}{\mu L}|\lambda_1 - \lambda_2|.$$

This demonstrates that $\boldsymbol{w}(\lambda)$ is Lipschitz continuous with constant $\frac{\sqrt{n}(L+\mu)}{\mu L}$. $\qquad\square$

PROOF OF SPARSITY CHARACTERIZATION

**Lemma** (Sparsity characterization). *Under Assumption 1, the following necessary and sufficient sparsity gauge holds:*

$$w_i(\lambda) = 0 \iff |\nabla_i f(\boldsymbol{w}_{-i}(\lambda))| \leq \lambda. \tag{6}$$

*Proof.* Let $\boldsymbol{w} = \boldsymbol{w}(\lambda)$ for some $\lambda \geq 0$. We fix an index $i \in \{1, \ldots, n\}$ and consider the function $h(t) = \zeta(t) + \lambda|t|$, where $\zeta(t) = f(w_1, \ldots, w_{i-1}, t, w_{i+1}, \ldots, w_n)$.

Since $f$ is $\mu$-strongly convex, $\zeta(t)$ is $\mu$-strongly convex in the scalar variable $t$ and the value $w_i$ must minimize $h(t)$ with respect to $t$.

We remind the reader on the assumption made at the beginning of Appendix B about $w_i \geq 0$. The first-order optimality condition for minimizing $\zeta(t)$ is $0 \in \partial h(w_i)$. We have $\zeta'(t) = \frac{\partial f}{\partial w_i}(w_1, \ldots, t, \ldots, w_n) = \nabla_i f(w_1, \ldots, t, \ldots, w_n)$.

The optimality condition becomes:

$$-\zeta'(w_i) \in \lambda\partial|w_i|$$

If $w_i = 0$, then $\partial|w_i| = [-1, 1]$. The condition becomes $\zeta_i'(0) \in [-\lambda, \lambda]$, which is equivalent to $|\zeta'(0)| = |\nabla_i f(x_{-i})| \leq \lambda$. This establishes: $w_i = 0 \implies |\nabla_i f(x_{-i})| \leq \lambda$.

Now we prove the converse: assume $|\nabla_i f(x_{-i})| = |\zeta'(0)| \leq \lambda$.

Since $\zeta$ is $\mu$-strongly convex, then its derivative is strictly increasing. Let's suppose, for contradiction, that $w_i > 0$, then $\zeta'(w_i) > f_i'(0) \geq -\lambda$. The optimality condition requires that $\zeta'(w_i) = -\lambda$, which contradicts the supposition. Therefore, $w_i = 0$.

Hence, $w_i = 0$ if and only if $|\zeta'(0)| = |\nabla_i f(x_{-i})| \leq \lambda$.

$\qquad\square$

### B.3   PROOF OF THEOREM 1

**Theorem 1** (Stable regularizations). *Let $\phi : \mathbb{R}^n \to \mathbb{R}_+$ be a continuous nonnegative function. Then there exists $\lambda \geq 0$ such that $\lambda = \phi \circ \boldsymbol{w}(\lambda)$. Define the compact set of $\phi$-stable regularizations w.r.t to $f$ as:*

$$\Lambda(\phi) := \{\lambda \geq 0 \,|\, \phi \circ \boldsymbol{w}(\lambda) = \lambda\}. \tag{7}$$

*Proof.* Define the function $g : \mathbb{R}_+ \to \mathbb{R}$ as $g(\lambda) = \phi(\boldsymbol{w}(\lambda)) - \lambda$. By Lemma 2, the mapping $\lambda \mapsto \boldsymbol{w}(\lambda)$ is continuous on its domain, and $\phi$ is continuous, hence their composition is continuous and so is $g$.

At $\lambda = 0$, $g(0) = \phi(\boldsymbol{w}(0)) - 0 \geq 0$, since $\phi$ is nonnegative.

Recall that $\boldsymbol{w}(\lambda) \in \mathcal{W}_\Lambda$ for all $\lambda$ and $\mathcal{W}_\Lambda$ is compact by Lemma 1. As $\phi$ is continuous on the compact set $\mathcal{W}_\Lambda$, it is bounded on $\mathcal{W}_\Lambda$, hence $M = \max_{\boldsymbol{w} \in \mathcal{W}_\Lambda} \phi(\boldsymbol{w}) < \infty$ is well-defined and finite. This gives the uniform bound $\phi(\boldsymbol{w}(\lambda)) \leq M$ for all $\lambda \geq 0$.

Consider the limit of $g(\lambda)$ as $\lambda \to \infty$: $g(\lambda) = \phi(\boldsymbol{w}(\lambda)) - \lambda \geq M - \lambda$. Thus, $\lim_{\lambda \to \infty} g(\lambda) = -\infty$.

To summarize, we have:

- $g$ is continuous on $\mathbb{R}_+$,

- $g(0) \geq 0$,

- $\lim_{\lambda \to \infty} g(\lambda) = -\infty$,

thus the Intermediate Value Theorem (IVT) guarantees that there exists at least one $\lambda_\star \in \mathbb{R}_+$ such that $g(\lambda_\star) = 0$, i.e., a value $\lambda_\star$ where $\phi(\boldsymbol{w}(\lambda_\star)) = \lambda_\star$.

Since $g$ is continuous and $\Lambda(\phi) = g_{-1}(0)$, the compactness of $\Lambda(\phi)$ follows. $\qquad\square$

### B.4   PROOF OF THEOREM 2

**Theorem 2** (Practical characterization). *Let $\boldsymbol{w} = \boldsymbol{w}(\lambda)$. For any $i \in [n]$, if $\exists \alpha > 0$ such that $|\nabla_i f(\boldsymbol{w}) - \alpha w_i| \leq \lambda$ then $w_i = 0$. Consequently, for any $\boldsymbol{\alpha} \in \mathbb{R}^n_{>0}$, the function $\psi_{\kappa,\boldsymbol{\alpha}} : \boldsymbol{w} \mapsto |\nabla f(\boldsymbol{w}) - \boldsymbol{\alpha} \odot \boldsymbol{w}|_{[k+1]}$, where $\odot$ is the element-wise product, satisfies:*

$$\lambda \in \Lambda(\psi_{\kappa,\boldsymbol{\alpha}}) \implies \|\boldsymbol{w}(\lambda)\|_0 \leq \kappa.$$

*Proof.* Assume we have such an $\alpha$, and we'll show $w_i$ must equal 0. Consider the case $w_i > 0$, then from the optimality conditions $\nabla f_i(x) = -\lambda$ is negative, and so is $-\alpha w_i$ so

$$-\lambda - \alpha w_i = -|\lambda + \alpha w_i| = -|-\lambda - \alpha w_i| = -|\nabla_i f(\boldsymbol{w}) - \alpha w_i| \geq -\lambda$$

implying $-\alpha w_i \geq 0$, a contradiction. Similarly if $w_i < 0$ then $\lambda - \alpha w_i = |\lambda - \alpha w_i| = |\nabla_i f(\boldsymbol{w}) - \alpha w_i| \leq \lambda$ implying $-\alpha w_i \leq 0$ i.e., $w_i \geq 0$, also a contradiction.

Hence the only remaining case is $w_i = 0$.

Consequently, the $(k+1)$-st largest magnitude in $\phi_\kappa$ or $\psi_\kappa$ is of an inactive component $i$ s.t $|\nabla_i f(\boldsymbol{w}_{-i})|$ and $|\nabla_i f(\boldsymbol{w}) - \alpha w_i|$ are equal (to $|\nabla_i f(\boldsymbol{w})|$). $\qquad\square$

### B.5   PROOF OF LEMMA 4 AND COROLLARY 1

**Lemma 4** (Decay Rate of $\delta_t$). *Let $\rho = \sqrt{1 - \mu/L}$ and $\eta \in (0, 1/L)$. The error $\delta_t$ satisfies:*

$$\delta_{t+1} \leq \rho(1 + \beta_t L_{\boldsymbol{w}} L_\psi)\delta_t + \rho L_{\boldsymbol{w}} \beta_t |\Phi(\lambda_t)| \tag{13}$$

*Proof.* $\boldsymbol{w}_{t+1}$ comes from one iteration of PGD for $\lambda_{t+1}$:

$$\begin{aligned}
\delta_{t+1} &\leq \rho\|\boldsymbol{w}_t - \boldsymbol{w}(\lambda_{t+1})\| && \text{using Equation (4)} \\
&\leq \rho(\|\boldsymbol{w}_t - \boldsymbol{w}(\lambda_t)\|_2 + \|\boldsymbol{w}(\lambda_t) - \boldsymbol{w}(\lambda_{t+1})\|_2) \\
&\leq \rho(\delta_t + L_{\boldsymbol{w}}|\lambda_{t+1} - \lambda_t|) && \text{using Lemma 3.}
\end{aligned}$$

From the iteration in Equation (10) we have:

$$|\lambda_{t+1} - \lambda_t| \le \beta_t |\psi(\boldsymbol{w}_t) - \lambda_t| \qquad \text{inequality due to non-expansiveness of } \Pi_{[0,\lambda_{\max}]}$$
$$\le \beta_t(\epsilon_t + |\Psi(\lambda_t) - \lambda_t|) \qquad\qquad \text{recall } \Psi(\lambda) = \psi(\boldsymbol{w}(\lambda))$$
$$\le \beta_t L_\psi \delta_t + \beta_t |\Phi(\lambda_t)|$$

Combining the two results:

$$\delta_{t+1} \le \rho(1 + \beta_t L_{\boldsymbol{w}} L_\psi)\delta_t + \rho L_{\boldsymbol{w}} \beta_t |\Phi(\lambda_t)|.$$

$\square$

**Corollary 1** (Boundedness of iterates)**.** *Let* $\overline{\beta} = \sup_t \beta_t$ *such that* $\rho(1 + \overline{\beta} L_{\boldsymbol{w}} L_\psi) < 1$*, then the sequence* $(\boldsymbol{w}_t)_t$ *generated by the iteration from Equation* (11) *is bounded, and* $\delta_t = O(\beta_t)$*.*

*Proof.* Define $\overline{\beta} := \sup_t \beta_t$ then:

$$\delta_{t+1} \le \rho\big(1 + \overline{\beta} L_{\boldsymbol{w}} L_\psi\big)\delta_t + \rho L_{\boldsymbol{w}} \overline{\beta} |\Phi(\lambda_t)|$$

Writing $q := \rho\big(1 + \overline{\beta} L_{\boldsymbol{w}} L_\psi\big)$ and $C_0 = \max_{\lambda \in [0,\lambda_{\max}]} L_{\boldsymbol{w}} \overline{\beta} |\Phi(\lambda)|$, we get:

$$\delta_{t+1} \le q\delta_t + C_0.$$

Hence, if $q < 1$, which is equivalent to $\overline{\beta} < \frac{\rho-1}{\rho L_{\boldsymbol{w}} L_\psi}$, then unrolling this recurrence from $i = 0$ to $t - 1$:

$$\delta_t \le q^t \delta_0 + C_0 \sum_{i=1}^{t-1} q^i \le q\delta_0 + \frac{C_0}{1-q}.$$

Thus $\boldsymbol{w}_t$ is always a bounded distance away from $\boldsymbol{w}(\lambda_t)$, and $\boldsymbol{w}(\lambda_t) \in \mathcal{W}_\Lambda$ and $\mathcal{W}_\Lambda$ was shown to be bounded in Lemma 1, hence all the $\boldsymbol{w}_t$ stay within a bounded set. $\square$

### B.6 PROOF OF THEOREM 3

*Proof.* We can write the $\lambda$-update as an update towards a root of $\Phi(\lambda) = \Psi(\lambda) - \lambda$:

$$\lambda_{t+1} = \lambda_t + \beta_t(\psi(\boldsymbol{w}_t) - \lambda_t)$$
$$= \lambda_t + \beta_t \left((\Psi(\lambda_t) - \lambda_t) + (\psi(\boldsymbol{w}_t) - \Psi(\lambda_t))\right)$$
$$= \lambda_t + \beta_t \Phi(\lambda_t) + \beta_t \epsilon_t,$$

where $\epsilon_t := \psi(\boldsymbol{w}_t) - \Psi(\lambda_t) = \psi(\boldsymbol{w}_t) - \psi(\boldsymbol{w}(\lambda_t))$.

The function $\psi$ is $L_\psi$-Lipschitz (Lemma 3). Thus,

$$|\epsilon_t| \le L_\psi \|\boldsymbol{w}_t - \boldsymbol{w}(\lambda_t)\|_2 = L_\psi \delta_t. \tag{24}$$

From Lemma 4, with $\beta_t = \beta_0/(t + t_0) = O(1/t)$, we have $\delta_t = O(\beta_t) = O(1/t)$. Therefore, $\epsilon_t = O(1/t)$.

Let $\Delta_t := |\lambda_t - \lambda_\star|$. The update for $\Delta_t$ is:

$$\Delta_{t+1} = |\lambda_{t+1} - \lambda_\star|$$
$$= |\lambda_t + \beta_t \Phi(\lambda_t) + \beta_t \epsilon_t - \lambda_\star|$$
$$= |(\lambda_t - \lambda_\star) + \beta_t(\Phi(\lambda_t) - \Phi(\lambda_\star)) + \beta_t \epsilon_t|,$$

since $\Phi(\lambda_\star) = \Psi(\lambda_\star) - \lambda_\star = 0$.

Consider the Lyapunov function $V_t = \frac{\Delta_t^2}{2} = \frac{1}{2}(\lambda_t - \lambda_\star)^2$.

$$V_{t+1} = \frac{1}{2}(\Delta_t + \beta_t(\Phi(\lambda_t) - \Phi(\lambda_\star)) + \beta_t \epsilon_t)^2$$
$$= V_t + \beta_t \Delta_t \left[\Phi(\lambda_t) - \Phi(\lambda_\star)\right] + \beta_t \Delta_t \epsilon_t + \frac{\beta_t^2}{2}(\Phi(\lambda_t) - \Phi(\lambda_\star) + \epsilon_t)^2.$$

$\Psi$ is $L_\Psi$-Lipschitz with $L_\Psi = L_\psi L_w$. Thus, $\Phi(\lambda) = \Psi(\lambda) - \lambda$ is $L_\Phi := (L_\Psi + 1)$-Lipschitz. Since $\lambda_\star$ is an asymptotically stable equilibrium for $\dot\lambda = \Phi(\lambda)$, for $\lambda_t$ in a neighborhood of $\lambda_\star$, we have:

$$\Delta_t(\Phi(\lambda_t) - \Phi(\lambda_\star)) = (\lambda_t - \lambda_\star)(\Phi(\lambda_t) - \Phi(\lambda_\star)) \le -c(\lambda_t - \lambda_\star)^2 = -cV_t.$$

The term $\beta_t \Delta_t \epsilon_t$ satisfies:

$$\beta_t \Delta_t \epsilon_t \le 2\beta_t \sqrt{V_t} L_\psi \delta_t.$$

The final term satisfies:

$$\frac{\beta_t^2}{2}(\Phi(\lambda_t) - \Phi(\lambda_\star) + \epsilon_t)^2 \le \beta_t^2\left[(\Phi(\lambda_t) - \Phi(\lambda_\star))^2 + |\epsilon_t|^2\right] \le \beta_t^2\left(L_\Phi|\Delta_t|^2 + |\epsilon_t|^2\right),$$

Since the algorithm ensures $\lambda_t$ remains bounded (as established in Lemma 4), this term is $O(\beta_t^2)$.

So we get:
$$V_{t+1} \le V_t - 2c\beta_t V_t + 2\beta_t \sqrt{V_t}(1 + L)\delta_t + O(\beta_t^2).$$

Using $\delta_t = O(\beta_t)$:

$$V_{t+1} \le (1 - 2c\beta_t + O(\beta_t^2))V_t + O(\beta_t^2\sqrt{V_t}) + O(\beta_t^2). \tag{25}$$

This form of recurrence is standard in stochastic approximation theory. Using Robbins-Siegmund theorem and its corollary (Liu & Yuan, 2024, Lemma 3), although we're operating in a deterministic case, for the rate $V_t = O(1/t)$ (or $\mathbb{E}[V_t] = O(1/t)$ in stochastic settings), one typically needs $2c\beta_0 > 1$. Then the terms $O(\beta_t^2\sqrt{V_t})$ and $O(\beta_t^2)$ are $O(t^{-2}\sqrt{V_t})$ and $O(t^{-2})$, so we get:

$$V_{t+1} \le (1 - \frac{2c\beta_0}{t} + O(\frac{1}{t^2}))V_t + O(\frac{1}{t^2}\sqrt{V_t}) + O(\frac{1}{t^2})$$

$$\le \left(1 - \frac{2c\beta_0}{t}\right)V_t + O(t^{-2.5}) + O(t^{-2})$$

For a self-contained proof, we can show by induction that if $\beta_0$ is large enough and $\beta_t = \frac{\beta_0}{t + t_0}$, then $V_t = O(1/t)$: assume that $V_t \le \frac{C}{t}$ for some $t > T$, then $V_{t+1} \le \frac{C}{t}(1 - \frac{2c\beta_0}{t}) + O(t^{5/2}) + O(\beta_t^2)$. Using Taylor expansion $\frac{1}{t+1} = \frac{1}{t} - \frac{1}{t^2} + o(t^{-3})$, we get $V_{t+1} \le \frac{C}{t+1}$ is equivalent to $2c\beta_0 > 1$.

If $2c\beta_0 > 1$, then for large $t$, $V_t = O(1/t)$. This gives $\Delta_t^2 = O(1/t)$.

Now we want to bound $\|\boldsymbol{w}_t - \boldsymbol{w}\|^2 = \|\boldsymbol{w}_t - \boldsymbol{w}(\lambda_\star)\|^2$. Using the triangle inequality:

$$\|\boldsymbol{w}_t - \boldsymbol{w}\|_2 \le \|\boldsymbol{w}_t - \boldsymbol{w}(\lambda_t)\|_2 + \|\boldsymbol{w}(\lambda_t) - \boldsymbol{w}(\lambda_\star)\|_2$$
$$\le \delta_t + L_w\Delta_t,$$

where $L_w$ is the Lipschitz constant of $\boldsymbol{w}(\cdot)$.

We have $\delta_t = O(\beta_t) = O(1/t)$, and $\Delta_t^2 = O(1/t)$. Thus, $\|\boldsymbol{w}_t - \boldsymbol{w}\| = O(1/t) + O(1/\sqrt{t}) = O(1/\sqrt{t})$. This establishes Equation (16) and completes the proof. $\qquad\square$

### B.7 CONVERGENCE OF SASTRA (THEOREM 4)

We provide a self-contained proof. The structure and the logic are similar to the deterministic case, although several of the arguments we use here are known results under various names, especially the notion of a two-timescale stochastic ODE from Borkar (2008).

Our goal is to prove that $\mathbb{E}[V_{t+1} \mid \mathcal{F}_t] \le (1 - C\eta_t)V_t + O(\eta_t)$ for some constant $C > 0$, which would guarantee that $\delta_t$ converges a.s. to 0 with $\mathbb{E}[\delta_t^2] = O(\eta_t)$.

**Lemma 5** (Tracking a moving target). *With $\eta_t \le 1/L$, define the tracker $\Delta_t = |\lambda_{t+1} - \lambda_t|$, the Lyapunov $V_t = \frac{1}{2}\delta_t^2 = \frac{1}{2}\|\boldsymbol{w}_t - \boldsymbol{w}(\lambda_t)\|^2$ obeys:*

$$\mathbb{E}[V_{t+1}|\mathcal{F}_t] \le (1 - \tfrac{\mu\eta_t}{2})V_t + \frac{L_{\boldsymbol{w}}^2}{\mu\eta_t}\mathbb{E}\left[\Delta_t^2|\mathcal{F}_t\right] + \tfrac{\sigma^2}{2}\eta_t^2 \tag{26}$$

*Proof.* Let $\mathcal{F}_t$ be the filtration up to time $t$. Using the non-expansiveness of the proximal operator and the contractive SGD step with $\eta_t \leq 1/L$ we get:

$$\mathbb{E}\left[V_{t+1}|\mathcal{F}_t\right] \leq \frac{1}{2}(1 - \mu\eta_t)\mathbb{E}\left[\|\boldsymbol{w}_t - \boldsymbol{w}(\lambda_{t+1})\|^2|\mathcal{F}_t\right] + \frac{1}{2}\eta_t^2\mathbb{E}\left[\|g_t - \nabla f(\boldsymbol{w}_t)\|^2|\mathcal{F}_t\right] \quad (27)$$

In order to get that $(1 - C\eta_t)$ factor, we use Young's inequality:

$$\|\boldsymbol{u} + \boldsymbol{v}\|^2 \leq (1 + \tau)\|\boldsymbol{u}\|^2 + (1 + \frac{1}{\tau})\|\boldsymbol{v}\|^2, \forall \tau > 0,$$

with $\boldsymbol{u} = \boldsymbol{w}_t - \boldsymbol{w}(\lambda_t)$, $\boldsymbol{v} = \boldsymbol{w}(\lambda_t) - \boldsymbol{w}(\lambda_{t+1})$, and $\tau = \frac{\mu\eta_t}{2(1-\mu\eta_t)}$ we get:

$$(1 - \mu\eta_t)(1 + \tau) = 1 - \frac{\mu\eta_t}{2}, \quad (1 - \mu\eta_t)\left(1 + \frac{1}{\tau}\right) \leq \frac{2}{\mu\eta_t}. \quad (28)$$

Plugging Equation (28) into Equation (27), with $\boldsymbol{v} \leq L_{\boldsymbol{w}}\Delta_t$ (from Lemma 2), we get:

$$\mathbb{E}\left[V_{t+1}|\mathcal{F}_t\right] \leq (1 - \frac{\mu\eta_t}{2})V_t + \frac{L_{\boldsymbol{w}}^2}{\mu\eta_t}\mathbb{E}\left[(\lambda_{t+1} - \lambda_t)^2|\mathcal{F}_t\right] + \frac{\sigma^2}{2}\eta_t^2 \quad (29)$$

$\square$

**Lemma 6** (Drift bound for $\Delta_t$). *Let $e_t := \lambda_t - \lambda_\star$ and $\delta_t = \|\boldsymbol{w}_t - \boldsymbol{w}(\lambda_t)\|$. Then, for a universal constant $C$:*

$$\mathbb{E}[\Delta_t^2 \mid \mathcal{F}_t] \leq C\beta_t^2\left(|\Phi(\lambda_t) - \lambda_t|^2 + L_\psi^2\delta_t^2 + \sigma_\psi^2\right). \quad (30)$$

*Proof.* Projection is 1-Lipschitz, so $\Delta_t \leq \beta_t|\psi_t - \lambda_t|$. Write $\psi_t = \psi(\boldsymbol{w}_t) + \xi_t = \Phi(\lambda_t) + (\psi(\boldsymbol{w}_t) - \Phi(\lambda_t)) + \xi_t$, use $|\psi(\boldsymbol{w}_t) - \Phi(\lambda_t)| \leq L_\psi\delta_t$, $(a+b+c)^2 \leq 3(a^2+b^2+c^2)$, and $\mathbb{E}[\xi_t^2 \mid \mathcal{F}_t] \leq \sigma_\psi^2$. $\square$

**Lemma 7** (State mean-square drift). *Under equation 15, there exist constants $C_1, C_2 > 0$ such that*

$$\mathbb{E}[e_{t+1}^2 \mid \mathcal{F}_t] \leq (1 - 2c\beta_t)e_t^2 + C_1\beta_t^2\delta_t^2 + C_2\beta_t^2 \quad (31)$$

*Consequently, for $\beta_t = \dfrac{\beta_0}{(t + t_0)(\log(t + t_0))^q}$ with any $q > 0$,*

$$\mathbb{E}[e_t^2] = O(\beta_t). \quad (32)$$

*Proof.* From the update:

$$e_{t+1} = e_t + \beta_t(\psi_t - \lambda_t) = e_t + \beta_t(\Phi(\lambda_t) - \lambda_t) + \beta_t b_t + \beta_t\xi_t, \quad b_t := \psi(\boldsymbol{w}_t) - \Phi(\lambda_t), |b_t| \leq L_\psi\delta_t.$$

Expand and take $\mathbb{E}[\cdot \mid \mathcal{F}_t]$:

$$\mathbb{E}[e_{t+1}^2 \mid \mathcal{F}_t] = e_t^2 + 2\beta_t e_t(\Phi(\lambda_t) - \lambda_t) + 2\beta_t e_t b_t + \beta_t^2\mathbb{E}[(\Phi(\lambda_t) - \lambda_t + b_t + \xi_t)^2 \mid \mathcal{F}_t].$$

Use $\mathbb{E}[\xi_t] = 0$, $\mathbb{E}[\xi_t^2] \leq \sigma_\psi^2$, Young's inequality $2e_t b_t \leq e_t^2 + b_t^2$, and the stability from Equation (15):

$$e_t(\Phi(\lambda_t) - \lambda_t) = e_t(\Phi(\lambda_t) - \Phi(\lambda_\star)) - e_t^2 \leq -ce_t^2 - e_t^2 = -(1 + c)e_t^2.$$

Therefore, the $2\beta_t$ term contributes at most $-2(1 + c)\beta_t e_t^2 + \beta_t e_t^2 + \beta_t b_t^2 = -(1 + 2c)\beta_t e_t^2 + \beta_t b_t^2$. Absorb the remaining $\beta_t^2$ square into $C_1\beta_t^2\delta_t^2 + C_2\beta_t^2$. Then a standard Robbins-Monro argument yields equation 32. $\square$

**Theorem 4** (Convergence with log-slowed $\beta_t$). *Let $\eta_t = \dfrac{\eta_0}{(t+t_0)}$ and:*

$$\beta_t = \frac{\beta_0}{(t + t_0)(\log(t + t_0))^q}, \qquad q > \frac{1}{2},$$

*then $\sum_t \beta_t^2/\eta_t < \infty$ and $\sum_t \eta_t^2 < \infty$. Then $(\boldsymbol{w}_t, \lambda_t)$ converges a.s with mean rates:*

$$\mathbb{E}\left[\|\boldsymbol{w}_t - \boldsymbol{w}(\lambda_t)\|^2\right] = O(1/t), \quad \mathbb{E}\left[|\lambda_t - \lambda_\star|^2\right] = O(\beta_t).$$

*Proof.* Combining Lemmas 5–7 and taking expectations gives

$$\mathbb{E}\left[V_{t+1}\right] \le \left(1 - \tfrac{\mu\eta_t}{2}\right)\mathbb{E}\left[V_t\right] + \underbrace{C_1\frac{\beta_t^2}{\eta_t}}_{:=\vartheta_t}\mathbb{E}\left[V_t\right] + \frac{C_2\beta_t^2}{\eta_t}\mathbb{E}\left[e_t^2\right] + \frac{C_3\beta_t^2}{\eta_t} + \tfrac{\sigma^2}{2}\eta_t^2 \qquad (33)$$

for constants $C_1, C_2, C_3$ depending on $L_\psi, L_{\boldsymbol{w}}, \mu$. Choose

$$\eta_t = \frac{\eta_0}{t + t_0}, \qquad \beta_t = \frac{\beta_0}{(t + t_0)(\log(t + t_0))^q}, \qquad q > \tfrac{1}{2}.$$

then $\sum_t \vartheta_t = \sum_t C_1\beta_t^2/\eta_t < \infty$ and $\sum_t \eta_t^2 < \infty$. Let $M_t := \prod_{s<t}(1 + \vartheta_s)$ (bounded since $\sum \vartheta_s < \infty$). Multiplying equation 33 by $M_{t+1}$ and applying Robbins–Siegmund with $a_t = \mu\eta_t/2$ and Lemma 7 (so $\frac{\beta_t^2}{\eta_t}\mathbb{E}[e_t^2] = O(\beta_t^3/\eta_t)$ is summable) yields:

$$\boxed{\mathbb{E}\left[\|\boldsymbol{w}_t - \boldsymbol{w}(\lambda_t)\|^2\right] = O(\eta_t), \qquad \mathbb{E}\left[(\lambda_t - \lambda_\star)^2\right] = O(\beta_t).}$$

Hence $\mathbb{E}\left[\|\boldsymbol{w}_t - \boldsymbol{w}(\lambda_t)\|^2\right] = O(1/t)$ and $\mathbb{E}\|\lambda_t - \lambda_\star\|^2 = \tilde{O}(1/t)$ (with the log factor from $\beta_t$).

In order to get the rate for $\|\boldsymbol{w}_t - \boldsymbol{w}_\star\|$, use the decomposition:

$$\|\boldsymbol{w}_t - \boldsymbol{w}_\star\| \le \|\boldsymbol{w}_t - \boldsymbol{w}(\lambda_t)\| + \|\boldsymbol{w}(\lambda_t) - \boldsymbol{w}(\lambda_\star)\| \le O(\eta_t) + L_{\boldsymbol{w}}|\lambda_t - \lambda_\star| \le O(\eta_t)$$

$\square$

**Why log-slowing is necessary here.** The Lyapunov recursion equation 33 requires:

$$\sum_{t=1}^{\infty} \frac{\beta_t^2}{\eta_t} < \infty$$

With $\eta_t = \eta_0/(t + t_0)$, choosing $\beta_t = \beta_0/((t + t_0)(\log(t + t_0))^q)$ and any $q > \tfrac{1}{2}$ gives $\sum_t \beta_t^2/\eta_t \sim \sum_t \frac{1}{t(\log t)^{2q}} < \infty$, while preserving Robbins–Monro ($\sum \beta_t = \infty$, $\sum \beta_t^2 < \infty$).

*The curious reader* can also the ponder possibility of using variance-reduction techniques by increasing the batch size $B_t$ used to estimate $\psi_t$ so that the summability condition is $\sum_t \frac{\beta_t}{\eta_t B_t} < \infty$, which is true when both $\eta_t$ and $\beta_t$ are $\Theta(t)$ as long as $B_t > O(t^a)$ for some $a > 0$.

## B.8 PROOF OF GROUP SPARSITY CHARACTERIZATION

**Theorem 5** (Practical characterization 2). *If $\exists \boldsymbol{A} \in \mathcal{S}_{++}^{|\boldsymbol{b}|}$ such that $\|\nabla_{\boldsymbol{b}} f(\boldsymbol{w}(\lambda)) - \boldsymbol{A}\boldsymbol{w}_{\boldsymbol{b}}(\lambda)\| \le \lambda$ then $\|\boldsymbol{w}_{\boldsymbol{b}}\| = 0$. Hence, it follows that for $\psi_\kappa^{(\boldsymbol{A})} : \boldsymbol{w} \mapsto [\boldsymbol{u}(\boldsymbol{w})]_{[\kappa+1]}$ with $\boldsymbol{u}_j(\boldsymbol{w}) = \|\nabla_{\boldsymbol{b}_j} f(\boldsymbol{w}) - \boldsymbol{A}_j\boldsymbol{w}_{\boldsymbol{b}_j}\|$ and $\boldsymbol{A} \in \mathcal{S}_{++}^{\mathcal{B}}$, if $\lambda \in \Lambda(\psi_\kappa^{(\boldsymbol{A})})$ then the solution $\boldsymbol{w}(\lambda)$ has at most $\kappa$ active blocks, i.e. $\mathrm{nnz}\left(\{\|\boldsymbol{w}_{\boldsymbol{b}_1}(\lambda)\|, \ldots, \|\boldsymbol{w}_{\boldsymbol{b}_p}(\lambda)\|\}\right) \le \kappa$.*

*Proof.* The optimality (KKT) conditions for $\boldsymbol{w}^\star \in \arg\min_{\boldsymbol{w}} F_{\mathcal{G}}(\cdot; \lambda)$ are

$$\boldsymbol{0}_n \in \nabla f(\boldsymbol{w}^\star) + \lambda\partial\Omega_{\mathcal{B}}(\boldsymbol{w}^\star). \qquad (34)$$

These specialize block-wise as follows:

$$\begin{cases} \|\nabla_{\boldsymbol{b}} f(\boldsymbol{w}^\star)\|_2 \le \lambda, & \text{if } \boldsymbol{w}_{\boldsymbol{b}}^\star = 0, \\ \nabla_{\boldsymbol{b}} f(\boldsymbol{w}^\star) = -\lambda\dfrac{\boldsymbol{w}_{\boldsymbol{b}}^\star}{\|\boldsymbol{w}_{\boldsymbol{b}}^\star\|_2}, & \text{if } \boldsymbol{w}_{\boldsymbol{b}}^\star \ne 0. \end{cases} \qquad (35)$$

Fix any $\boldsymbol{A} \in S_{++}^{\mathcal{B}}$ and define the per-block nonnegative scores:

$$u_j(x) := \|\nabla_{\boldsymbol{b}_j} f(\boldsymbol{w}) - \boldsymbol{A}_j \boldsymbol{w}_{\boldsymbol{b}_j}\|_2, \qquad j = 1, \ldots, p, \qquad (36)$$

and the order-statistic functional

$$\psi_k^{(\boldsymbol{A})}(\boldsymbol{w}) := [u_1(\boldsymbol{w}), \ldots, u_p(\boldsymbol{w})]_{[k+1]} \in \mathbb{R}_+. \qquad (37)$$

Fix a block $\boldsymbol{b} = \boldsymbol{b}_j$. If $\boldsymbol{w}_{\boldsymbol{b}}^{\star} = 0$, there is nothing to prove. Suppose instead that $\boldsymbol{w}_{\boldsymbol{b}}^{\star} \neq 0$. By the block-wise KKT condition equation 35,

$$\nabla_{\boldsymbol{b}} f(\boldsymbol{w}^{\star}) = -\lambda \frac{\boldsymbol{w}_{\boldsymbol{b}}^{\star}}{\|\boldsymbol{w}_{\boldsymbol{b}}^{\star}\|_2}.$$

Write $\boldsymbol{w}_{\boldsymbol{b}}^{\star} = r\boldsymbol{v}$ with $r := \|\boldsymbol{w}_{\boldsymbol{b}}^{\star}\|2 > 0$ and $\|\boldsymbol{v}\|_2 = 1$. Then

$$\nabla_{\boldsymbol{b}} f(\boldsymbol{w}^{\star}) - \boldsymbol{A}_j \boldsymbol{w}_{\boldsymbol{b}}^{\star} = -\lambda \boldsymbol{v} - r\boldsymbol{A}_j \boldsymbol{v}.$$

Let $\gamma := \boldsymbol{v}^{\top} \boldsymbol{A}_j \boldsymbol{v} > 0$ (since $\boldsymbol{A}_i \succ 0$) and decompose $\boldsymbol{A}_j \boldsymbol{v} = \gamma \boldsymbol{v} + \boldsymbol{n}$ with $\boldsymbol{v} \perp \boldsymbol{n}$. Then

$$\|-\lambda \boldsymbol{v} - r\boldsymbol{A}_j \boldsymbol{v}\|_2^2 = \|(-\lambda - r\gamma)\boldsymbol{v} - r\boldsymbol{n}\|_2^2 = (\lambda + r\gamma)^2 + r^2 \|\boldsymbol{n}\|_2^2 \geq (\lambda + r\gamma)^2 > \lambda^2,$$

because $r\gamma > 0$. Hence $\|\nabla_{\boldsymbol{b}} f(\boldsymbol{w}^{\star}) - \boldsymbol{A}_i \boldsymbol{w}_{\boldsymbol{b}}^{\star}\|_2 > \lambda$. Take the contrapositive to yield the claim. $\square$

## C    EXTENDED NOTES

### C.1    RELAXING THE STRONG CONVEXITY ASSUMPTION

**Theorem C.1.** *Fix $\lambda > 0$ and let $\gamma > 0$. Define $\boldsymbol{w}_{\gamma} := \arg\min_{\boldsymbol{w}} F_{\gamma}(\boldsymbol{w})$ where the $\gamma$-strongly convex Tikhonov-perturbed objective is:*

$$F_{\gamma}(\boldsymbol{w}) := F(\boldsymbol{w}, \lambda) + \frac{\gamma}{2}\|\boldsymbol{w}\|^2,$$

*Let $\boldsymbol{w}^{\dagger} = \arg\min_{\boldsymbol{w} \in \boldsymbol{w}(\lambda)} \|w\|$ be the minimum-Euclidean-norm solution. Then, for every $\gamma > 0$ the following holds:*

- ***Norm shrinkage.*** $\|\boldsymbol{w}_{\gamma}\| \leq \|\boldsymbol{w}_{\dagger}\|$.

- ***Minimal-norm selection.*** $\boldsymbol{w}_{\gamma} \to \boldsymbol{w}^{\dagger}$ *as* $\gamma \to 0^+$.

- ***Exact value-gap identity and bound.***
$$0 \leq F(\boldsymbol{w}_{\gamma}, \lambda) - F(\boldsymbol{w}^{\dagger}, \lambda) \leq \frac{\gamma}{2}\left(\|\boldsymbol{w}^{\dagger}\|^2 - \|\boldsymbol{w}_{\gamma}\|^2\right) \tag{38}$$

*Proof.* We have the optimality conditions:
$$\boldsymbol{0}_n \in \partial F(\boldsymbol{w}_{\gamma}, \lambda) + \gamma \boldsymbol{w}_{\gamma}, \quad \text{and} \quad \boldsymbol{0}_n \in \partial F(\boldsymbol{w}^{\dagger}, \lambda). \tag{39}$$
Pick $g_{\gamma} \in \partial F(\boldsymbol{w}_{\gamma}, \lambda)$ and $g^{\star} \in \partial F(\boldsymbol{w}^{\dagger}, \lambda)$ with $g_{\gamma} = -\gamma \boldsymbol{w}_{\gamma}$ and $g^{\star} = 0$. Maximal monotonicity from Equation (23) yields:
$$\langle g_{\gamma} - g^{\star}, \boldsymbol{w}_{\gamma} - \boldsymbol{w}^{\dagger}\rangle \geq 0 \implies -\gamma \langle \boldsymbol{w}_{\gamma}, \boldsymbol{w}_{\gamma} - \boldsymbol{w}^{\dagger}\rangle \geq 0.$$
Hence, $\langle \boldsymbol{w}_{\gamma}, \boldsymbol{w}^{\dagger}\rangle \geq \|\boldsymbol{w}_{\gamma}\|^2$, and by Cauchy–Schwarz $\|\boldsymbol{w}_{\gamma}\| \leq \|\boldsymbol{w}^{\dagger}\|$.

For any $\boldsymbol{w}$,
$$F_{\gamma}(\boldsymbol{w}) \geq F_{\gamma}(\boldsymbol{w}_{\gamma}).$$
With $\boldsymbol{w} = \boldsymbol{w}^{\dagger}$:

$$F(\boldsymbol{w}_{\gamma}, \lambda) - F(\boldsymbol{w}^{\dagger}, \lambda) \leq \frac{\gamma}{2}\left(\|\boldsymbol{w}^{\dagger}\|^2 - \|\boldsymbol{w}_{\gamma}\|^2\right) \tag{40}$$

we also have:
$$F(\boldsymbol{w}_{\gamma}, \lambda) - F(\boldsymbol{w}^{\dagger}, \lambda) \geq 0 \tag{41}$$

From Equations (40) and (41) we get Equation (38).

Using the strict convexity of $F_{\gamma}$, an equivalent polarized form yields:
$$F(\boldsymbol{w}_{\gamma}, \lambda) - F(\boldsymbol{w}^{\dagger}, \lambda) \leq \gamma \langle \boldsymbol{w}_{\gamma}, \boldsymbol{w}^{\dagger} - \boldsymbol{w}_{\gamma}\rangle,$$

Let $\gamma_k \downarrow 0$ and $\boldsymbol{w}_{\gamma_k} \to \bar{\boldsymbol{w}}$ along a convergent subsequence (existence by compactness). Choose $g_{\gamma_k} \in \partial F(\boldsymbol{w}_{\gamma_k}, \lambda)$ with $g_{\gamma_k} = -\gamma_k \boldsymbol{w}_{\gamma_k} \to 0$. Closedness of the graph of $\partial F$ yields $0 \in \partial F(\bar{\boldsymbol{w}})$, so $\bar{\boldsymbol{w}} \in \boldsymbol{w}\lambda$. Moreover, the polarized inequality gives:
$$\langle \boldsymbol{w}_{\gamma_k}, \boldsymbol{w}^{\dagger} - \boldsymbol{w}_{\gamma_k}\rangle \geq 0 \implies \langle \bar{\boldsymbol{w}}, \boldsymbol{w}^{\dagger} - \bar{\boldsymbol{w}}\rangle \geq 0.$$
This characterizes the Euclidean projection of $\boldsymbol{0}_n$ onto the closed convex set $\boldsymbol{w}(\lambda)$; hence $\bar{\boldsymbol{w}} = P_{\boldsymbol{w}\lambda}(0) = \boldsymbol{w}^{\dagger}$. As every subsequence has the same limit $\boldsymbol{w}^{\dagger}$, the full sequence satisfies $\boldsymbol{w}_{\gamma} \to w^{\dagger}$ as $\gamma \downarrow 0$. $\square$

## C.2 DETERMINISTIC LINEAR RATE VIA INTERMITTENT SCHEME

**Theorem 6** (Linear Convergence Rate (Deterministic)). *Let $T$ be a fixed interval length, and assume $\Psi$ is contractive in a neighborhood of $\lambda_\star$, i.e., there exists a constant $\gamma \in [0, 1)$ such that $|\Psi(\lambda) - \Psi(\lambda_\star)| \leq \gamma|\lambda - \lambda_\star|$ for all $\lambda$ in this neighborhood. Consider the sequence $(\lambda_k)_k$ generated by the intermittent schedule $\beta \in (0, 1)$ and let $\lambda_k = \lambda_{kT}$. Then the sequence $(\lambda_k)_k$ converges linearly to a neighborhood of $\lambda_\star$ with a radius of $O(\rho^T)$. Specifically, the following bound holds:*

$$|\lambda_{k+1} - \lambda_\star| \leq \left[1 - \beta(1 - \gamma)\right]|\lambda_k - \lambda_\star| + O\left(\frac{\rho^T}{1 - \sigma}\right). \tag{42}$$

*Proof.* We analyze the intermittent schedule with constant $\beta$. Let $\lambda_k = \lambda_{kT}$ and $\boldsymbol{w}_k = \boldsymbol{w}_{kT}$.

In the interval $t \in [kT, (k+1)T - 1)$, $\lambda = \lambda_k$ is fixed. The $\boldsymbol{w}$-updates converge geometrically towards $\boldsymbol{w}(\lambda_k)$: $\|\boldsymbol{w}_{(k+1)T} - \boldsymbol{w}(\lambda_k)\| \leq \rho^T\|\boldsymbol{w}_{kT} - \boldsymbol{w}(\lambda_k)\|$. Let $D$ be an upper bound on $\|\boldsymbol{w}_t - \boldsymbol{w}(\lambda_k)\|$.

The error term in the $\lambda$-update at step $k + 1$ (using $\boldsymbol{w}_{(k+1)T}$) is $\epsilon_k = \psi_\kappa(\boldsymbol{w}_{(k+1)T}) - \Psi(\lambda_k)$. We have $|\epsilon_k| = |\psi_\kappa(\boldsymbol{w}_{(k+1)T}) - \psi_\kappa(\boldsymbol{w}(\lambda_k))| \leq L_\psi\|\boldsymbol{w}_{(k+1)T} - \boldsymbol{w}(\lambda_k)\| \leq L_\psi\rho^T\|\boldsymbol{w}_{kT} - \boldsymbol{w}(\lambda_k)\| \leq L_\psi D\rho^T$. This error can be made small by choosing $T$ large.

For the $\lambda$-update: $\lambda_{k+1} = \lambda_k + \beta(\Psi(\lambda_k) - \lambda_k + \epsilon_k)$. We have:

$$\begin{aligned}\Delta_{k+1} &= \Delta_k + \beta(\Psi(\lambda_k) - \Psi(\lambda_\star)) - \beta(\lambda_k - \lambda_\star) + \beta\epsilon_k \\ &= (1 - \beta)\Delta_k + \beta(\Psi(\lambda_k) - \Psi(\lambda_\star)) + \beta\epsilon_k\end{aligned}$$

Using the contractivity assumption, $|\Psi(\lambda_k) - \Psi(\lambda_\star)| \leq \gamma|\Delta_k|$:

$$\begin{aligned}|\Delta_{k+1}| &\leq |1 - \beta||\Delta_k| + \beta\gamma|\Delta_k| + \beta|\epsilon_k| \\ &\leq \underbrace{(|1 - \beta| + \beta\gamma)}_{\sigma}|\Delta_k| + \beta L_\psi D\rho^T\end{aligned}$$

For $0 < \beta \leq 1$, this requires $\sigma = 1 - \beta + \beta\gamma < 1$, which simplifies to $\beta(1 - \gamma) > 0$ (true since $\beta > 0, \gamma < 1$). Thus, $\sigma = 1 - \beta(1 - \gamma) < 1$.

The recurrence $|\Delta_{k+1}| \leq \sigma|\Delta_k| + C\rho^T$ (where $C = \beta L_\psi D$) shows linear convergence to a neighborhood of $\lambda_\star$. Specifically:

$$|\Delta_k| \leq \sigma^k|\Delta_0| + \frac{C\rho^T}{1 - \sigma} \tag{43}$$

The error converges linearly towards a ball of radius $O(\rho^T)$. For convergence to $\lambda_\star$, $T$ must be large enough relative to the desired precision. The convergence rate is $\sigma = 1 - \beta(1 - \gamma)$ per $k$ (per $T$ steps).

For the sequence $\boldsymbol{w}_t$:

$$\begin{aligned}\|\boldsymbol{w}_{kT} - \boldsymbol{w}\| &\leq \|\boldsymbol{w}_{kT} - \boldsymbol{w}(\lambda_k)\| + \|\boldsymbol{w}(\lambda_k) - \boldsymbol{w}(\lambda_\star)\| \\ &\leq \rho^T D' + L_h|\lambda_k - \lambda_\star|\end{aligned}$$

(where $D'$ bounds $\|\boldsymbol{w}_{kT} - \boldsymbol{w}(\lambda_k)\|$, which might depend on $k$). Both terms decrease towards zero (the first geometrically in $T$, the second linearly in $k$). $\qquad\square$

**Note on intermittent schedule:** If $\beta_t$ is zero except at $t = kT$, the tracking argument $\|\boldsymbol{w}_t - \boldsymbol{w}(\lambda_t)\| \to 0$ is strengthened. Between $\lambda$-updates, $\boldsymbol{w}_t$ converges geometrically towards the fixed $\boldsymbol{w}(\lambda_{kT})$. Thus, at times $t = kT$, the error $\|\boldsymbol{w}_{kT} - \boldsymbol{w}(\lambda_{(k-1)T})\|$ (or $\|\boldsymbol{w}_{kT} - \boldsymbol{w}(\lambda_{kT})\|$ depending on timing) can be made very small by choosing $T$ large, making the noise term $\epsilon_{kT}$ in the $\lambda$-update small. The overall ODE analysis structure remains applicable.

### C.3 SPARSITY ALGEBRA

A novel sparsity algebra is proposed to provide a structured and unified framework for applying diverse sparsity constraints to (neural network) parameters. This algebra is composed of three fundamental components: the *sparsity block*, the *sparsity group*, and *Group coupling*. These components can be combined to express a wide range of structured sparsity patterns, from fine-grained N:M sparsity to coarse-grained channel and layer-level pruning.

#### C.3.1 FORMULATION

Let a parameter tensor be denoted by $\mathbf{W} \in \mathbb{R}^{d_1 \times \cdots \times d_m}$. The proposed sparsity algebra introduces a hierarchical structure onto this tensor.

**Sparsity Block** The fundamental unit of sparsity is the *sparsity block*. The tensor $\mathbf{W}$ is partitioned into non-overlapping, equally-sized blocks. The block dimensions are defined by a tuple $b = (b_1, \ldots, b_m)$, where each tensor dimension $d_i$ must be divisible by the corresponding block dimension $b_i$. This creates a grid of blocks $\mathcal{B}$ with dimensions $B = (\frac{d_1}{b_1}, \ldots, \frac{d_m}{b_m})$. Each block $\mathbf{W}_{i_1, \ldots, i_m} \in \mathbb{R}^{b_1 \times \cdots \times b_m}$ consists of the elements of $\mathbf{W}$ indexed by $[i_1 b_1 : (i_1 + 1)b_1, \ldots, i_m b_m : (i_m + 1)b_m]$. The pruning decision is applied at the block level, meaning all weights within a single block are pruned or kept together.

**Sparsity Group** A *sparsity group* is a collection of sparsity blocks. The grid of blocks $\mathcal{B}$ is further partitioned into groups defined by a tuple $g = (g_1, \ldots, g_m)$, where for each axis $i$, the block grid dimension $B_i = d_i/b_i$ must be divisible by $g_i$. This results in a grid of groups $\mathcal{G}$ with dimensions $(\frac{B_1}{g_1}, \ldots, \frac{B_m}{g_m})$. Each group contains $|g| := \prod_{i=1}^{m} g_i$ sparsity blocks. The sparsity level $\kappa$ is applied within each group, resulting in a sparsity of $1 - \frac{\kappa}{|g|}$.

**Group Coupling** To enforce sparsity constraints jointly across multiple parameter tensors, the concept of *Group coupling* is introduced. A set of parameter tensors $\{\mathbf{W}^{(1)}, \ldots, \mathbf{W}^{(k)}\}$ can be coupled, allowing a global sparsity constraint to be applied across their corresponding groups. This is particularly useful for maintaining a global level of sparsity or for co-pruning related parameters. The coupling mechanism aligns the group grids of the different tensors, potentially with permutations of their axes, and applies a unified sparsity criterion across them.

#### C.3.2 SPARSITY EXAMPLES

The flexibility of this algebra allows for the formulation of various structured sparsity patterns.

**2:4 Sparsity** A common hardware-accelerated pattern is 2:4 sparsity, where two out of every four elements must be zero. This can be formulated using our algebra on a weight matrix $W \in \mathbb{R}^{m \times n}$. We first define a block size of $b = (1, 1)$ to consider individual weights. We then group these blocks into $1 \times 4$ contiguous sets by defining a group of size $g = (1, 4)$. Within each group, we enforce that only $\kappa = 2$ elements (blocks) are kept, thereby achieving 2:4 sparsity along the rows.

**$\kappa$-Block Row Sparsity** To enforce a fixed number of non-zero blocks in each row of a block-partitioned matrix, we can define a group that encompasses an entire row of blocks. For a matrix with block grid dimensions $(B_1, B_2)$, we set the group size to $g = (1, B_2)$. Within each of these row-groups, we can then keep $\kappa$ of blocks per row block (group).

**Channel-wise Sparsity** For a 2D convolutional layer, the weight tensor is typically of shape $\mathbf{W} \in \mathbb{R}^{c_{out} \times c_{in} \times k_h \times k_w}$, where $c_{out}$ and $c_{in}$ are the number of output and input channels, and $k_h, k_w$ are the kernel height and width. To achieve channel-wise sparsity where each output channel connects to only $\kappa$ input channels, define the blocks to be the filter weights corresponding to a single input-output channel connection, i.e., $b = (1, 1, k_h, k_w)$. The resulting block grid has dimensions $(c_{out}, c_{in}, 1, 1)$. We then group all blocks corresponding to a single output channel by setting the group size to $g = (1, c_{in}, 1, 1)$. For each of these groups, we enforce that only $\kappa$ blocks (i.e., connections to $\kappa$ input channels) are kept.

**Global Thresholding with Coupling** A significant challenge in pruning is determining the appropriate sparsity level for each layer. Global thresholding applies a single criterion across the entire network, imposing a uniform sparsity pressure without requiring per-layer hyperparameter tuning. Our algebra provides a structured mechanism for this via group coupling.

For instance, consider an MLP with two layers, $\boldsymbol{W}_1 \in \mathbb{R}^{1024 \times 2048}$ and $\boldsymbol{W}_2 \in \mathbb{R}^{2048 \times 1024}$. We can prune entire neurons in the hidden layer by coupling the columns of $\boldsymbol{W}_1$ with the rows of $\boldsymbol{W}_2$. We set a block size of $b^{(1)} = b^{(2)} = (16, 16)$. The block grids are $B^{(1)} = (64, 128)$ and $B^{(2)} = (128, 64)$. To group along the hidden dimension, we set group sizes $g^{(1)} = (64, 1)$ and $g^{(2)} = (1, 64)$. These yields group grids of shape $(1, 128)$ for $\boldsymbol{W}_1$ and $(128, 1)$ for $\boldsymbol{W}_2$. By coupling these two group grids (permuting the axes of one to align them), we can jointly select the top $\kappa$ groups, effectively keeping only $\kappa \times 16$ neurons in the hidden layer.

Alternatively, for global block-level sparsity, we can set the group size equal to the block grid size for each tensor ($g = B$). This creates one large group per tensor. Coupling these groups allows us to keep $\kappa$ blocks across the entire 2-layer MLP.

## C.4 DENSE- 4:16 BLOCK-SPARSE MATMUL KERNEL

```python
import torch
import triton
import triton.language as tl

"""
Grouped Block Sparse GEMM Benchmark
===================================
Pattern: For every GROUP_SIZE (=16) contiguous 16x16 blocks along the K
    dimension,
only NNZ_PER_GROUP (2, 4, 8) blocks are  n o n zero . This is a structured
    block sparsity
pattern analogous to extended N:M at a 16x16 block granularity.

Storage Layout:
  Indices  : [num_col_blocks, k_group_count, NNZ_PER_GROUP]
  Values   : [num_col_blocks, k_group_count, NNZ_PER_GROUP, B_K, B_N]
Flattened in  r o w major  order (last dimension fastest). The kernel
    expects both tensors
flattened to 1-D contiguous buffers with the above logical order.

Kernel Mapping:
  Each program instance (pid_m, pid_n) computes a tile of C of shape
  (BLOCK_SIZE_M, B_N). It iterates over all K groups and, inside each
      group,
  iterates the NNZ_PER_GROUP  n o n zero  16x16 blocks, performing one dot
      per block.

Limitations / Assumptions:
  - K must be divisible by (GROUP_SIZE * B_K)
  - N must be divisible by B_N
  - Block size B_K x B_N = 16 x 16
"""

DTYPE = torch.float16

@triton.jit
def grouped_block_sparse_kernel_vec(
    a_ptr,
    b_values_ptr,
    b_indices_ptr,
    c_ptr,
    M,
```

```
1404        N,
1405        K,
1406        stride_am,
1407        stride_ak,
1408        stride_cm,
1409        stride_cn,
1410        B_K: tl.constexpr,
1411        B_N: tl.constexpr,
1412        GROUP_SIZE: tl.constexpr,  # 16
1413        NNZ_PER_GROUP: tl.constexpr,  # 2,4,8
1414        BLOCK_SIZE_M: tl.constexpr,
1415        GROUP_M: tl.constexpr,
1416    ):
1417        """
1418        For each group g:
1419            1. Load NNZ_PER_GROUP local block indices.
1420            2. Map to global block indices (add g * GROUP_SIZE).
1421            3. Gather all (NNZ_PER_GROUP * B_K) columns from A into a single
1422                tile.
1423            4. Load contiguous B values segment for the group.
1424            5. Perform one tl.dot and accumulate.
1425        This reduces loop overhead inside the group and enables a larger
1426            fused dot.
1427        """
1428        pid_m = tl.program_id(axis=0)
1429        pid_n = tl.program_id(axis=1)
1430        num_pid_m = tl.cdiv(M, BLOCK_SIZE_M)
1431        num_pid_n = tl.cdiv(N, B_N)
1432
1433        pid_m, pid_n = tl.swizzle2d(pid_m, pid_n, num_pid_m, num_pid_n,
1434            GROUP_M)
1435
1436        m_start = pid_m * BLOCK_SIZE_M
1437        n_start = pid_n * B_N
1438        offs_m = m_start + tl.arange(0, BLOCK_SIZE_M)
1439        offs_n = n_start + tl.arange(0, B_N)
1440        c_ptrs = c_ptr + offs_m[:, None] * stride_cm + offs_n[None, :] *
1441            stride_cn
1442        c_mask = (offs_m[:, None] < M) & (offs_n[None, :] < N)
1443        accumulator = tl.zeros((BLOCK_SIZE_M, B_N), dtype=tl.float32)
1444
1445        k_blocks = K // B_K
1446        groups_per_col = k_blocks // GROUP_SIZE
1447        indices_per_col = groups_per_col * NNZ_PER_GROUP
1448        values_per_col = indices_per_col * B_K * B_N
1449        col_indices_base = b_indices_ptr + pid_n * indices_per_col
1450        col_values_base = b_values_ptr + pid_n * values_per_col
1451        offs_i = tl.arange(0, NNZ_PER_GROUP)
1452        b_offs = tl.arange(0, NNZ_PER_GROUP * B_K * B_N)
1453        group_indices_ptr = col_indices_base
1454        b_group_start = col_values_base
1455        b_ptrs = b_group_start + b_offs
1456
1457        for g in range(groups_per_col):
1458            b_group_flat = tl.load(b_ptrs)

            local_block_indices = tl.load(group_indices_ptr + offs_i)
            global_block_indices = g * GROUP_SIZE + local_block_indices
            k_offsets_scattered = (
                global_block_indices[:, None] * B_K + tl.arange(0, B_K)[None,
                    :]
            )
            k_offsets_flat = tl.reshape(k_offsets_scattered, (NNZ_PER_GROUP *
                B_K,))
            a_ptrs = (
```

```
            a_ptr
            + offs_m[:, None] * stride_am
            + k_offsets_flat[None, :] * stride_ak
        )
        a_tile = tl.load(a_ptrs, mask=(offs_m[:, None] < M), other=0.0)

        b_tile = tl.reshape(b_group_flat, (NNZ_PER_GROUP * B_K, B_N))
        accumulator += tl.dot(a_tile, b_tile)

        # b_group_start += NNZ_PER_GROUP * (B_K * B_N)
        group_indices_ptr += NNZ_PER_GROUP
        b_ptrs += NNZ_PER_GROUP * (B_K * B_N)
    tl.store(c_ptrs, accumulator, mask=c_mask)
```

## C.5 PROXIMAL ADAM OPTIMIZER

| **Algorithm 3** Adam Optimizer | **Algorithm 4** Group Proximal Adam |
|---|---|
| **Require:** Step size $\eta$, decay rates $\beta_1, \beta_2 \in [0,1)$, objective $f(\boldsymbol{w})$ | **Require:** Regularization $\Omega_{\mathcal{B},\mathcal{G}}$ and Adam optimizer. |
| 1: **Initialize** $m_0 \leftarrow 0, v_0 \leftarrow 0, t \leftarrow 0$ | **Initialize** Adam$(\alpha, \beta_1, \beta_2)$ |
| 2: **while** $t < T$ **do** | **while** $t < T$ **do** |
| 3: $\quad t \leftarrow t + 1$ | $\quad$ Run one Adam step to get $\boldsymbol{v} = \boldsymbol{w}_t$ |
| 4: $\quad g_t \leftarrow \nabla_\theta f_t(\boldsymbol{w}_{t-1})$ | $\quad$ Define $\boldsymbol{H} = \mathrm{diag}(\hat{v}_t)$ |
| 5: $\quad m_t \leftarrow \beta_1 \cdot m_{t-1} + (1 - \beta_1) \cdot g_t$ | $\quad$ **for** each block $b \in \mathcal{B}$ **do** |
| 6: $\quad v_t \leftarrow \beta_2 \cdot v_{t-1} + (1 - \beta_2) \cdot g_t^2$ | $\quad\quad$ **if** $\|\boldsymbol{H}_{b,b}\boldsymbol{v}_b\| \le \eta\lambda$ **then** |
| 7: $\quad \widehat{m}_t \leftarrow m_t / (1 - \beta_1^t)$ | $\quad\quad\quad \boldsymbol{w}_{t,b} \leftarrow \boldsymbol{0}.$ |
| 8: $\quad \widehat{v}_t \leftarrow v_t / (1 - \beta_2^t)$ | $\quad\quad$ **else** |
| 9: $\quad \boldsymbol{w}_t \leftarrow \boldsymbol{w}_{t-1} - \alpha \cdot \widehat{m}_t / (\sqrt{\widehat{v}_t} + \epsilon)$ | $\quad\quad\quad$ Solve for $\mu$ using Equation (45) |
| 10: **end while** | $\quad\quad\quad$ Set $\boldsymbol{w}_b \leftarrow (\boldsymbol{H}_{b,b} + \mu\boldsymbol{I})^{-1}\boldsymbol{H}_b\boldsymbol{v}_b$ |
| 11: **return** $\boldsymbol{w}_T$ | $\quad\quad$ **end if** |
| | $\quad$ **end for** |
| | **end while** |
| | **return** $\boldsymbol{w}_T$ |

To extend the soft-thresholding operator to the class of adaptive gradient methods, we construct a local quadratic approximation of the objective $F$ at iteration $t$. We employ the norm $\|\cdot\|_{\boldsymbol{H}_t}$ induced by the symmetric positive-definite preconditioner $\boldsymbol{H}_t$, defined such that $\langle \boldsymbol{x}, \boldsymbol{y} \rangle_{\boldsymbol{H}_t} = \langle \boldsymbol{x}, \boldsymbol{H}_t \boldsymbol{y} \rangle$.

The surrogate objective function is given by:

$$F_t(\boldsymbol{w}) \approx f(\boldsymbol{w}_t) + \langle \hat{\boldsymbol{m}}_t, \boldsymbol{w} - \boldsymbol{w}_t \rangle + \frac{1}{2\eta_t}\|\boldsymbol{w} - \boldsymbol{w}_t\|_{\boldsymbol{H}_t}^2 + \Omega_{\mathcal{B},\mathcal{G}}(\boldsymbol{w}),$$

where $\hat{\boldsymbol{m}}_t$ denotes the first moment estimate of the gradient.

The update rule is the minimizer of this regularized quadratic model:

$$\boldsymbol{w}_{t+1} = \mathrm{prox}_{\eta_t\Omega}^{\boldsymbol{H}_t}\left(\boldsymbol{w}_t - \eta_t\boldsymbol{H}_t^{-1}\hat{\boldsymbol{m}}_t\right), \tag{44}$$

where the proximal operator is defined under the $\boldsymbol{H}_t$-norm as:

$$\mathrm{prox}_{\eta_t\Omega}^{\boldsymbol{H}_t}(\mathbf{u}) \triangleq \arg\min_{\boldsymbol{w}} \left( \frac{1}{2\eta_t}\|\boldsymbol{w} - \mathbf{u}\|_{\boldsymbol{H}_t}^2 + \Omega_{\mathcal{B},\mathcal{G}}(\boldsymbol{w}) \right).$$

When $\Omega_{\mathcal{B},\mathcal{G}}$ is the element-wise $\ell_1$ regularization, we obtain the same Proximal Adam expression introduced by Melchior et al. (2020), which is equivalent to using a pseudo element-wise learning rate $\hat{\eta}_t = \frac{\eta_t}{\boldsymbol{H}_{ii}}$.

**Proposition 1** (Separable Adaptive Soft-Thresholding with Adam). *Let $\Omega(\boldsymbol{w}) = \lambda\|\boldsymbol{w}\|_1$ and let $\boldsymbol{H}_t = \mathrm{diag}(\mathbf{h}_t)$ be a diagonal positive-definite matrix. Given the intermediate iterate $\mathbf{u}_t = \boldsymbol{w}_t - \eta_t\boldsymbol{H}_t^{-1}\hat{\boldsymbol{m}}_t$, the update $\boldsymbol{w}_{t+1}$ is given element-wise by:*

$$w_{t+1,i} = \mathrm{sgn}(\mathbf{u}_{t,i}) \cdot \max\left(0, |u_{t,i}| - \frac{\eta_t\lambda}{\mathbf{h}_{t,i}}\right).$$

*Proof.* Since $\boldsymbol{H}_t$ is diagonal and the $\ell_1$ norm is separable, the objective $\Phi(\boldsymbol{w})$ decouples into independent scalar problems for each coordinate $i$. Dropping the subscript $t$ for brevity:

$$\min_{w_i} \left\{ \frac{h_i}{2\eta}(w_i - u_i)^2 + \lambda|w_i| \right\}.$$

The first-order optimality condition requires zero to be in the subdifferential of the objective with respect to $w_i$:

$$0 \in \frac{h_i}{\eta}(w_i - u_i) + \lambda\partial|w_i|.$$

Rearranging for $u_i$:

$$u_i \in w_i + \frac{\eta\lambda}{h_i}\partial|w_i|.$$

which equivalent to the Euclidean soft-thresholding with $\hat\lambda = \frac{\lambda}{h_i}$. Hence:

$$w_i = \operatorname{sgn}(u_i)\max\left(0, |u_i| - \frac{\eta\lambda}{h_i}\right).$$

$\square$

With group $\ell_1$ regularization $\Omega_{\mathcal{B},\mathcal{G}}$, there is no closed form for the proximal operator, but we can compute it efficiently using a root finding method (Becker et al., 2018; Adler et al., 2020).

**Adam with Block-wise soft-thresholding**    With a diagonal $\boldsymbol{H}_t$, the problem decouples to a soft threshold per-block using the group threshold $\lambda := \lambda_i$ for some $i \in \{1\ldots, q\}$. We denote the intermediate update for block $b$ as:

$$\boldsymbol{v} = \boldsymbol{w}_{b,t} - \eta_t \boldsymbol{H}_{t,b}^{-1}\hat{\boldsymbol{m}}_{b,t}$$

Without loss of generality, we will drop the block index $b$ and assume we have a single block:

$$\boldsymbol{w}^* = \arg\min_{\boldsymbol{w}} \frac{1}{2\eta_t}\|\boldsymbol{w} - \boldsymbol{v}\|_{\boldsymbol{H}_t}^2 + \lambda\|\boldsymbol{w}\|_2$$

then we need to find $\boldsymbol{w}^*$ such that:

$$-\frac{1}{\eta_t}\boldsymbol{H}(\boldsymbol{w}^* - \boldsymbol{v}) \in \lambda\partial\|\boldsymbol{w}^*\|_2$$

It's is straightforward to show that:

$$\|\boldsymbol{H}\boldsymbol{v}\| \le \eta_t\lambda \iff \boldsymbol{w}^* = \boldsymbol{0}$$

We notice here that the Euclidean metric ($\boldsymbol{H} = \boldsymbol{I}$) is a special case.

If $\|\boldsymbol{H}\boldsymbol{v}\| > \eta_t\lambda$ then $\boldsymbol{w}^* \ne \boldsymbol{0}$ and $\partial\|\boldsymbol{w}^*\|_2 = \frac{\boldsymbol{w}^*}{\|\boldsymbol{w}^*\|_2}$ and we get:

$$\frac{1}{\eta_t}\boldsymbol{H}(\boldsymbol{w}^* - \boldsymbol{v}) + \lambda\frac{\boldsymbol{w}^*}{\|\boldsymbol{w}^*\|_2} = \boldsymbol{0}$$

Denote $\mu := \frac{\eta_t\lambda}{\|\boldsymbol{w}^*\|} > 0$, then:

$$\boldsymbol{H}(\boldsymbol{w}^* - \boldsymbol{v}) + \mu\boldsymbol{w}^* = \boldsymbol{0}$$

The solution is then:

$$\boldsymbol{w}^* = (\boldsymbol{H} + \mu I)^{-1}\boldsymbol{H}\boldsymbol{v} = \left[\frac{H_{ii}}{H_{ii} + \mu}\right]_{i \in b} \circ \boldsymbol{v}$$

The scalar $\mu > 0$ is the unique root of the one-dimensional equation for $\boldsymbol{H} = \operatorname{diag}(H_{ii})$:

$$\eta_t\lambda = \zeta(\mu) = \mu\|(\boldsymbol{H} + \mu I)^{-1}\boldsymbol{H}\boldsymbol{v}\|_2 = \mu\sqrt{\sum_{i \in b}\left(\frac{H_{ii}}{H_{ii} + \mu}v_i\right)^2} \tag{45}$$

$\zeta$ is strictly increasing in $\mu$ with $\zeta(0) = 0$ and $\zeta(+\infty) = \|\boldsymbol{H}\boldsymbol{v}\| > \eta_t\lambda$, so the equation has a unique solution. It can be found efficiently by a simple bisection search.

The group soft-thresholding operator for a diagonal Newton matrix is therefore:

$$S_{\eta\lambda,\boldsymbol{H}}(\boldsymbol{v}) = \begin{cases} 0, & \text{if } \|\boldsymbol{H}\boldsymbol{v}\|_2 \le \eta\lambda \\ (\boldsymbol{H} + \mu I)^{-1}\boldsymbol{H}\boldsymbol{v}, & \text{otherwise,} \end{cases}$$

where $\mu > 0$ solves Equation (45). In component form:

$$\boldsymbol{w}_{t+1} = \begin{cases} 0, & i \notin G, \\ \dfrac{\boldsymbol{H}_{ii}}{\boldsymbol{H}_{ii} + \mu}\, \boldsymbol{v}_i, & i \in G. \end{cases}$$

When the Hessian is the identity ($H_k = I$), $h_i = 1$ for all $i$, it reduces to $\eta\lambda = \mu \|\boldsymbol{v}\|_2/(1+\mu)$, yielding the familiar block soft-thresholding $\left(1 - \frac{\eta\lambda}{\|\boldsymbol{v}\|_2}\right)_+ \boldsymbol{v}$.

Let:

$$S := \|\boldsymbol{H}\boldsymbol{v}\|_2^2, \qquad h_{\min} := \min_i \boldsymbol{H}_{ii}, \qquad m_{\max} := \max_i \boldsymbol{H}_{ii}.$$

We bound the sum by min/max of $h_{ii}$, because every term in the sum is positive:

$$\frac{1}{(h_{\max} + \mu)^2} \ \leq\ \frac{1}{(\boldsymbol{H}_{ii} + \mu)^2} \ \leq\ \frac{1}{(h_{\min} + \mu)^2},$$

Multiplying by the non-negative constants $\boldsymbol{H}_{ii}^2\boldsymbol{v}_i^2$ and summing gives

$$S\,\frac{\mu^2}{(h_{\max} + \mu)^2} \ \leq\ \lambda^2 \ \leq\ S\,\frac{\mu^2}{(h_{\min} + \mu)^2}.$$

Both sides increase monotonically in $\mu$. Therefore, the solution $\mu^*$ is bracketed by the two solutions of the equalities:

$$S\,\frac{\mu^2}{(h_{\max} + \mu)^2} \ =\ \lambda^2, \qquad S\,\frac{\mu^2}{(h_{\min} + \mu)^2} \ =\ \lambda^2.$$

Hence, provided that the solution exists (i.e. $\sqrt{S} > \lambda$), the unique positive root is:

$$\boxed{\mu(h) \ =\ \frac{\lambda\,h}{\sqrt{S} - \lambda}}$$

(the denominator is the same for both $h_{\min}$ and $h_{\max}$).

With the two extreme values of $h$ we obtain:

$$\boxed{\mu_{\text{low}} \ =\ \frac{\lambda\,h_{\min}}{\sqrt{S} - \lambda}, \qquad \mu_{\text{high}} \ =\ \frac{\lambda\,h_{\max}}{\sqrt{S} - \lambda}}$$

and

$$\boxed{\mu_{\text{low}} \ \leq\ \mu^* \ \leq\ \mu_{\text{high}}}$$

These bounds are inexpensive to compute (one pass to obtain $S, h_{\min}, h_{\max}$) and provide a tight interval that guarantees convergence of the Newton step for the soft-thresholding operator in a convex optimization problem.

Early exit: if $\eta\lambda \geq \sqrt{S}$ (equivalently $\|H_b\boldsymbol{v}_b\|_2 \leq \eta\lambda$) we immediately set the block component to zero and skip all further work.

**Proximal AdamW**   To derive the AdamW optimization step within the local linearization framework, we must first understand why standard $L_2$ regularization fails to yield the desired decoupled weight decay.

Recall the local quadratic approximation for standard Adam at time $t$, using the norm induced by $\boldsymbol{H}_t = \mathrm{diag}(\hat{v}_t)^{1/2}$:

$$\min_{\boldsymbol{w}} \left[ f(\boldsymbol{w}_t) + \langle \boldsymbol{w} - \boldsymbol{w}_t, \hat{\boldsymbol{m}}_t \rangle + \frac{1}{2\eta_t} \|\boldsymbol{w} - \boldsymbol{w}_t\|_{\boldsymbol{H}_t}^2 \right] \tag{46}$$

Taking the gradient with respect to $\boldsymbol{w}$ and setting it to zero yields the standard Adam update:

$$\hat{\boldsymbol{m}}_t + \frac{1}{\eta_t} \boldsymbol{H}_t (\boldsymbol{w} - \boldsymbol{w}_t) = 0 \implies \boldsymbol{w}_{t+1} = \boldsymbol{w}_t - \eta_t \boldsymbol{H}_t^{-1} \hat{\boldsymbol{m}}_t$$

If we add standard $L_2$ regularization, $\frac{\rho}{2}\|\boldsymbol{w}\|_2^2$, to the objective in Eq. equation 46, the gradient condition becomes:

$$\hat{\boldsymbol{m}}_t + \rho \boldsymbol{w} + \frac{1}{\eta_t} \boldsymbol{H}_t (\boldsymbol{w} - \boldsymbol{w}_t) = 0$$

Assuming the approximation $\boldsymbol{w} \approx \boldsymbol{w}_t$ for the regularization gradient, the update becomes:

$$\boldsymbol{w}_{t+1} \approx \boldsymbol{w}_t - \eta_t \boldsymbol{H}_t^{-1} (\hat{\boldsymbol{m}}_t + \rho \boldsymbol{w}_t)$$

Here, the decay term $\rho \boldsymbol{w}_t$ is scaled by the preconditioner $\boldsymbol{H}_t^{-1}$. This couples the regularization with the adaptive learning rates, leading to inconsistent decay across parameters—the issue AdamW aims to solve.

ADAMW FORMULATION

AdamW decouples weight decay from the adaptive gradient step. The target update rule is:

$$\boldsymbol{w}_{t+1} = \boldsymbol{w}_t - \eta_t \boldsymbol{H}_t^{-1} \hat{\boldsymbol{m}}_t - \eta_t \rho \boldsymbol{w}_t$$

To achieve this within the proximal framework, we must adjust the regularization term in the objective function to counteract the geometry of $\boldsymbol{H}_t$.

We introduce a **preconditioned linearization** of the regularization term. Instead of the standard Euclidean inner product, we align the penalty with the metric $\boldsymbol{H}_t$:

$$\Omega_t(\boldsymbol{w}) = \rho \langle \boldsymbol{w} - \boldsymbol{w}_t, \boldsymbol{w}_t \rangle_{\boldsymbol{H}_t}$$

Substituting this into the local approximation objective:

$$f_t^{\mathrm{AdamW}}(\boldsymbol{w}) = \underbrace{f(\boldsymbol{w}_t) + \langle \boldsymbol{w} - \boldsymbol{w}_t, \hat{\boldsymbol{m}}_t \rangle}_{\text{Linearized Loss}} + \underbrace{\rho \langle \boldsymbol{w} - \boldsymbol{w}_t, \boldsymbol{w}_t \rangle_{\boldsymbol{H}_t}}_{\text{Preconditioned Decay}} + \underbrace{\frac{1}{2\eta_t} \|\boldsymbol{w} - \boldsymbol{w}_t\|_{\boldsymbol{H}_t}^2}_{\text{Trust Region}} \tag{47}$$

We minimize $f_t^{\mathrm{AdamW}}(\boldsymbol{w})$ with respect to $\boldsymbol{w}$. Recall that $\langle \boldsymbol{x}, \boldsymbol{y} \rangle_{\boldsymbol{H}_t} = \boldsymbol{x}^\top \boldsymbol{H}_t \boldsymbol{y}$.

$$\nabla_{\boldsymbol{w}} f_t^{\mathrm{AdamW}}(\boldsymbol{w}) = \hat{\boldsymbol{m}}_t + \rho \boldsymbol{H}_t \boldsymbol{w}_t + \frac{1}{\eta_t} \boldsymbol{H}_t (\boldsymbol{w} - \boldsymbol{w}_t)$$

Setting the gradient to zero:

$$0 = \hat{\boldsymbol{m}}_t + \rho \boldsymbol{H}_t \boldsymbol{w}_t + \frac{1}{\eta_t} \boldsymbol{H}_t (\boldsymbol{w} - \boldsymbol{w}_t)$$

$$-\frac{1}{\eta_t} \boldsymbol{H}_t (\boldsymbol{w} - \boldsymbol{w}_t) = \hat{\boldsymbol{m}}_t + \rho \boldsymbol{H}_t \boldsymbol{w}_t$$

Multiplying both sides by $-\eta_t \boldsymbol{H}_t^{-1}$:

$$\boldsymbol{w} - \boldsymbol{w}_t = -\eta_t \boldsymbol{H}_t^{-1} \hat{\boldsymbol{m}}_t - \eta_t \rho \underbrace{\boldsymbol{H}_t^{-1} \boldsymbol{H}_t}_{I} \boldsymbol{w}_t$$

$$\boldsymbol{w}_{t+1} = \boldsymbol{w}_t - \eta_t \left( \boldsymbol{H}_t^{-1} \hat{\boldsymbol{m}}_t + \rho \boldsymbol{w}_t \right)$$

This recovers the exact AdamW update step, where the weight decay $\rho \boldsymbol{w}_t$ is applied isotropically, independent of the adaptive scaling $\boldsymbol{H}_t^{-1}$.

As such, the proximal AdamW is equivalent to proximal Adam if we use the update $\boldsymbol{v} = \boldsymbol{w}_t - \eta_t(\boldsymbol{H}_t^{-1} \hat{\boldsymbol{m}}_t + \rho \boldsymbol{w}_t)$

