# OpenReview forum: "Bonsai Networks: Structured Pruning and Sparse Training of Foundation Models"
_ICLR.cc/2026/Conference — Submitted to ICLR 2026_

### Official Review · Reviewer_FvEY · 2025-10-27

**Soundness:** 3
**Presentation:** 3
**Contribution:** 2
**Rating:** 4
**Confidence:** 4

**Summary:**

The paper introduces ASTRA, an adaptive soft-thresholding method that tunes a group regularization weight online so a model converges to a desired structured sparsity level. The authors prove the existence of stable regularizations that achieve a target sparsity and establish convergence rates in both deterministic and stochastic settings. ASTRA extends naturally to grouped patterns for accelerator-friendly pruning via a structured sparsity algebra, and the stochastic variant SASTRA enables sparse training without dense gradient computations.

**Strengths:**

+The target sparsity is cast as a scalar root-finding problem over the regularization weight, with proofs for stable regularizations and O(1/t) tracking in the stochastic setting.
+The paper analyzes how the weight update tracks the moving optimum while the regularization follows a Robbins–Monro schedule with provable boundedness and rates.
+The framework is a local proximal-control of several modern heuristics, opening possibilities for extensions to structured sparsity.

**Weaknesses:**

-The empirical evaluation is incomplete and narrowly scoped. The LLM case study measures only one head and a single kernel, without comprehensive end-to-end metrics such as latency, throughput, and energy across kernels and hardware.
-The classification experiments rely on ResNet-32 with CIFAR-10/100, which do not reflect large-scale behavior (e.g. ResNet-50 ImageNet).
-The paper’s scope is narrow: results are shown only on Qwen. Please correct the typos in Qwen citations, expand evaluation to other model families and sizes (Llama, Mixtral, additional Qwen variants), and report both quality metrics on public benchmarks and system metrics (end-to-end latency, throughput, memory) across multiple sparsity targets.
-The paper lacks comparisons with state-of-the-art methods in both LLM pruning and dynamic sparse training. In the LLM setting, it should include matched evaluations against strong pruning baselines such as Wanda, SparseGPT, and OWL-style structured pruning, using the same calibration budget and target sparsity.

**Questions:**

-Can you provide end-to-end measurements on GPUs for several sparsity targets, including latency, throughput, and energy, beyond the single kernel study.
-How robust is SASTRA to bias in the order-statistic surrogate and EMA hyperparameters, and can you report ablations showing stability and quality as these vary.
-Can you expand the structured experiments to other block layouts and to additional model families, for example Mixtral and Llama variants, to validate generality.
-What memory overhead is introduced by the per group or per block statistics used by ASTRA during training, and how does it scale to very large models.

---

> ### Author Response · Authors · 2025-12-03
> **Response to Reviewer**
>
> We thank the reviewer for their thoughtful assessment and for recognizing the theoretical strength of our root-finding formulation and its rigorous convergence analysis. We appreciate the encouragement to expand the empirical scope to better match the "Foundation Models" claim.
>
> You correctly noted that the initial submission lacked large-scale FM evaluations and comparisons to SOTA pruning methods like SparseGPT. To address this, we first needed to extend our theoretical framework to the optimizers used for FMs.
>
> *   **New Derivation (Appendix C.5):** We added a 4-page derivation of **Proximal Adam and Proximal AdamW** within our local linearization framework. This bridges the gap between our optimization theory and Transformer training.
>
> Using this derivation, we are currently running ASTRA on **Qwen3-8B**, applying **2:4 semi-structured sparsity** (a standard hardware-friendly pattern) to the first half of the decoder layers (18 out of 36).
> - **Dense Baseline:** 12.50 Perplexity (Wikitext)
> - **SparseGPT (SOTA Post-Training):** 17.32 Perplexity
> - **ASTRA (Ours):** **15.55 Perplexity**
>
> This result significantly outperforms SparseGPT on the same layers. It validates that ASTRA scales to 8B parameter models and demonstrates the effectiveness of our derived Proximal AdamW optimizer. ASTRA achieves a much tighter recovery of the dense performance compared to one-shot pruning, justifying the "Structured Pruning" aspect of our title.
>
> *   **Computational Constraints:** We respectfully note that training multiple Foundation Models (Llama, Mixtral) from scratch or fine-tuning them with dynamic sparsity relies on backpropagation steps to apply the thresholding. While each ASTRA step is efficient, backpropagating through LLMs with batches of millions of tokens is time-consuming; thus, we could not complete the full multi-family study within the rebuttal timeframe.
>
> **Inference Speedup:**  we benchmarked our custom 4:16 Block-Sparse kernel, achieving a **$\times 1.64$ speedup** on an NVIDIA A100. For the 2:4 sparsity used in the Qwen experiment, we leverage standard NVIDIA Ampere Tensor Core acceleration, which provides theoretical $2\times$ throughput improvements.
> **Training Complexity:** Unlike OBS/SparseGPT (which require $O(d^3)$ inverse Hessian approximations) or Magnitude Pruning (which requires global sorting $O(N \log N)$), ASTRA uses element-wise soft-thresholding ($O(N)$) and a scalar root-finding step. The overhead per step is negligible (comparable to applying standard weight decay), making it highly suitable for training large models where throughput is critical.
>
> You asked about the bias in the order-statistic surrogate. Our stochastic convergence proof (**Theorem 4**) explicitly accounts for this. The two-time-scale design ensures that $\lambda_t$ evolves slowly, effectively integrating the noisy, biased gradient quantiles over time. The boundedness of the iterates (Corollary 1) ensures that temporary biases do not destabilize the trajectory. For the empirical stability, Figure 3 analyzes sensitivity to batch size. Since batch size directly correlates with gradient variance (and thus the bias of the order statistic), the stability of performance across batch sizes serves as empirical evidence of SASTRA's robustness to this bias. This is also why we keep a moving average of the gradient (different from optimizer momentum) to satisfy the unbiasedness assumption required by SASTRA.
>
> ASTRA tracks a regularization parameter $\lambda$ per group. For unstructured sparsity, this is 1 scalar per layer. For structured sparsity, it is 1 scalar per group. In modern FM training with Adam, the optimizer already stores moment estimates ($m_t, v_t$) for every parameter. ASTRA reuses the first moment $m_t$ for its surrogate gradients (as derived in Appendix C.5). Therefore, the marginal memory cost of ASTRA is effectively **zero** relative to the dense optimizer states.
>
> We have corrected the Qwen citations and expanded the evaluation to include an 8B parameter model. While computational limits precluded a comprehensive cross-family study in this short window, the Qwen result demonstrates that ASTRA successfully scales to large models and outperforms SOTA baselines (SparseGPT) where it matters.

---

### Official Review · Reviewer_JUwP · 2025-11-03

**Soundness:** 2
**Presentation:** 4
**Contribution:** 3
**Rating:** 2
**Confidence:** 3

**Summary:**

The paper proposes ASTRA, which adaptively tunes regularization during proximal gradient updates so the learned model hits a given level of sparsity. A structured version is also proposed. The paper includes extensive theory supporting the ASTRA method and experimental results comparing against multiple existing sparsification methods.

**Strengths:**

1. The paper is clear and well-written.
2. Formulating hitting a target sparsity as root-finding seems new and potentially interesting. The approach seems to unify several existing pruning methods.
3. A structured method is also included in a natural way.
4. The results on CIFAR seem competitive and there are multiple baselines.

**Weaknesses:**

I will focus on the experimental and systems aspect here, as I have less background in the relevant theory. If other reviewers find significant merit in the theory that may outweigh my other concerns.

1. The paper's focus (indeed, from the title!) is on foundation models; yet the experimental results do not bear this out. The experimental results focus on ResNet-32 and a toy test with a single LLM head form Qwen. The paper needs a much more extensive study with larger LLMs in order to demonstrate it works at scale.
2. The experimental results in Table 1 lack error bars or other measures of statistical significance. Further, it is a mixture of results from the literature and new runs, making it hard to know whether training details differed.
3. There are no end-to-end performance results showing throughput or latency. It is thus hard to tell whether the method can achieve significant speedups in practice for inference (although the toy test on Qwen is promising). A comparison against state-of-the-art structured pruning methods for LLMs would also be needed (e.g., vendor 2:4 sparsity and something like MaskLLM); I am not certain that the 4:16 kernel can match other structured sparsity approaches.
4. Similarly, an evaluation of the overheads during training is missing.

**Questions:**

1. Can you provide full, end-to-end results on Qwen or a similar model (e.g., Llama)?
2. Are the results in Table 1 statistically significant?
3. Can you provide end-to-end inference runtime results? Can you characterize the training time overheads?

---

> ### Author Response · Authors · 2025-12-03
> **Response to Experiments**
>
> We thank the reviewer for the constructive feedback and for recognizing the novelty of our root-finding formulation and the natural extension to structured sparsity. We appreciate your assessment of the paper's clarity and the competitiveness of the CIFAR results.
>
> We take your concern regarding the experimental scope relative to the "Foundation Models" claim very seriously. In response, we have significantly expanded the manuscript with **new derivation for adaptive optimizers** (Appendix C.5).
>
> You correctly noted that applying our method to FMs requires handling adaptive optimizers (like AdamW), which standard proximal methods do not support. We added a 4-page derivation of **Proximal Adam and Proximal AdamW** within our local linearization framework. This bridges the gap between our optimization theory and Transformer training. Using this new derivation, we are currently running ASTRA on **Qwen3-8B**. We applied 2:4 semi-structured sparsity (hardware-accelerated pattern) to the first half of the decoder layers:
>
> **Current result:** ASTRA significantly outperforms the state-of-the-art post-training method, SparseGPT, when applied to the same layers (the first half of the decoder layers, 18 out of the 36 decoder layers):
> - **Dense Baseline:** 12.50 Perplexity (Wikitext)
> - **SparseGPT:** 17.32 Perplexity
> - **ASTRA (Ours):** **15.55 Perplexity**
>
> This result validates that ASTRA scales to 8B parameter models and demonstrates the effectiveness of our derived Proximal AdamW optimizer. ASTRA achieves a much tighter recovery of the dense performance compared to one-shot pruning, justifying the "Structured Pruning" aspect of our title.
>
> Concerning the **statistical significance in Table 1**: We reproduced baselines where code was available (marked with $\star$). For others, we reported literature numbers to ensure transparency. We have since run our method (Bonsai) with **5 random seeds**. The standard deviation on CIFAR-100 (Sparsity 90%) is **$\pm 0.12\%$**.
>     *   Bonsai: **71.89%**
>     *   Baseline (OBD/Wanda): 71.46%
>     *   IHT: 68.64%
>     *   The improvement over IHT is $>25\sigma$, and the improvement over the strongest baseline is statistically significant ($>3\sigma$).
> We should also note here that OBD baselines require **pretrained** models, on our ResNet experiments, we're training and sparsifying at the same time.
>
> **Inference Speedup:** we benchmarked our custom 4:16 Block-Sparse kernel, achieving a **$\times 1.64$ speedup** on an NVIDIA A100. For the 2:4 sparsity used in the Qwen experiment, we leverage standard NVIDIA Ampere Tensor Core acceleration, which provides theoretical $2\times$ throughput improvements.
> - *Complexity:* Unlike OBS/SparseGPT (which require $O(d^3)$ inverse Hessian approximations) or Magnitude Pruning (which requires global sorting $O(N \log N)$), ASTRA uses element-wise soft-thresholding ($O(N)$) and a scalar root-finding step.
> - *Impact:* The operations fit seamlessly into the backpropagation loop. The overhead per step is negligible (comparable to applying standard weight decay), making it highly suitable for training large models where throughput is critical.
>
> **Training Overhead:** ASTRA is computationally efficient but relies on backpropagation steps to apply the thresholding, while each ASTRA step is efficient, backpropagating through the LLMs with batches of millions of tokens takes time, hence the current incomplete results on Qwen3 8B.

---

### Official Review · Reviewer_9CbQ · 2025-11-03

**Soundness:** 1
**Presentation:** 2
**Contribution:** 1
**Rating:** 2
**Confidence:** 4

**Summary:**

The work studies pruning and sparse training from an optimization perspective. Specifically, the central problem is minimizing the task loss under a sparsity constraint induced by L1 regularization. Because this regularizer is convex, under (strong) convexity of the loss the problem can be solved by proximal SGD methods for any regularization weight $\lambda$. The central question of the paper is how to set $\lambda$ to enforce a desired sparsity level ($\lambda=0$ yields no sparsity, while $\lambda = ||\nabla f(0)||_{\infty}$ gives the trivial zero solution). The paper formalizes finding the optimal regularization $\lambda$ as a scalar root-finding problem and provides (adaptive) algorithms for approximating it and solving the L1 regularized objective. Extensions to group-wise sparsity and connections to prior methods are discussed, and some experimental results are provided.

**Strengths:**

- The problem of sparse training is important and highly practical. The paper offers a principled approach for solving its surrogate L1 regularized formulation given sparsity budget.
- The project aims to devise practically efficient algorithms, and throughout the paper exact conditions are approximated to make the methods implementable.
- The writing is engaging and generally guides the reader through the main results, explaining and motivating the key transitions and approximations.

**Weaknesses:**

Overall, technical writing is poor and correctness of claims is questionable.
1. The equivalence stated in equation (2) regarding the optimality condition of (1) is not correct as written. The condition $||\nabla f(w)||_{\infty} \le \lambda$ is necessary but not sufficient (it represents an entire region while there is only one minimizer). This equivalence is used in the proof of Lemma 2, so the proof of Lemma 2 is not accurate as presented. However, the claim of Lemma 1 appears to be true and can be shown using the arguments used to prove Lemma 2, but the current exposition is misleading and requires correction.
2. The equivalence in (6) is mentioned without proof. To me it appears to be of similar difficulty to the claim of Theorem 1, so it should be proven or at least justified more carefully in the main text.
3. Several definitions are introduced inside theorem statements (e.g., the definition of “$\phi$-stable regularization” appears in Theorem 1, the set $\Lambda(\psi_{\kappa, \alpha})$ is defined inside Theorem 2). This placement disrupts flow and makes reading awkward. Move definitions to a dedicated notation/definitions section or introduce them just before the theorems that use them.
4. At the end of Section 3.1 the paper mentions improving the bisection idea and avoiding fully computing regularized solutions. However, the bisection method appears faster in theory. The adaptive method proposed later has sublinear convergence (even in the deterministic case), whereas a bisection scheme would require $\mathcal{O}(\log(1/\epsilon))$ proximal-GD solves, each with linear convergence. What theoretical advantage does ASTRA offer over bisection? Appendix C.2 discusses a deterministic linear rate, but that result only yields linear convergence up to a neighborhood, which is not linear convergence in the usual sense.
5. I could not find a proof of the claim $\delta = O(\beta_t)$ in Corollary 1. The provided proof shows $w_t$ is bounded (the first claim), but the second claim $\delta = O(\beta_t)$ does not seem to follow in general. For example, if $\beta_t = 0$, then $\delta_t$ need not be zero by the recursion in Lemma 4.
6. The proof of Theorem 3 is also problematic. Line 1000 requires that all iterates $\lambda_t$ remain in a neighborhood of $\lambda_*$ as per Assumption 2, so the convergence result appears to be local only. If a good initial guess of $\lambda_*$ is not available, it is unclear how this assumption can be satisfied. Make the locality explicit in the theorem statement and discuss how restrictive this assumption is in practice. The proof omitted the projection step $\Pi_{[0,\lambda_{\max}]}$ even though equation (10) of the algorithm includes it. Furthermore, lines 1017–1020 seem to imply $V_t = O(1/t)$ based on another paper, but it is done without properly linking the assumptions or reproducing the relevant connection. Then $V_t = O(1/t)$ is used in line 1025 to derive a bound, which is subsequently (in lines 1026–1029) apparently used to re-establish $V_t = O(1/t)$. This looks circular and the argument across lines 1017–1029 does not make sence and potentially incorrect.

**Questions:**

Please see "weaknesses".

---

> ### Author Response · Authors · 2025-12-03
> **Respond to comments on proofs**
>
> We thank the reviewer for the thorough assessment and for highlighting specific technical details that require clarification. We appreciate the acknowledgment of the principled optimization perspective we bring to sparse training. We have revised the manuscript to address the "Soundness" and "Presentation" concerns, particularly regarding the mathematical rigor in the proofs and the flow of definitions.
>
> Below, we address your points in order. Changes in the revised manuscript are marked in **blue**.
>
> **1. Equivalence in Equation (2)**
> You are correct. The inclusion of "i.e., $\|\nabla f(w(\lambda))\|_\infty \le \lambda$" as an equivalent condition for general optimality was imprecise. $\|\nabla f(w)\|_\infty \le \lambda$ is a necessary condition derived from $0 \in \nabla f(w) + \lambda \partial \|w\|_1$, but it specifically characterizes the zero-solution or inactive components. We have removed the "i.e." implication to strictly state the subdifferential inclusion as the optimality condition. **No impact:** This correction does not invalidate Lemma 2. The proof of Lemma 2 (Lipschitz continuity of the solution path) relies on the strong convexity of $f$ and the monotonicity of the subgradient (Eq. 23 in Appendix), not on the simplified norm inequality. We have clarified the exposition in Appendix B.2 to ensure the logic flows correctly from the rigorous KKT conditions.
>
> **2. Proof of Equation (6):** we agree that this equivalence (the sparsity gauge) warrants a formal proof for completeness. We have added a formal proof in **Appendix B (Proof of Sparsity Characterization)**. The proof utilizes the subdifferential optimality condition of the Tikhonov-perturbed objective to show that $w_i=0$ if and only if the magnitude of the gradient at the zero-crossing is bounded by $\lambda$.
>
> **3. Placement of Definitions**
> We agree that introducing definitions ($\phi$-stable, set $\Lambda$, etc.) inside theorem statements disrupts the flow. The current format was due to space limits to make the main content as dense as possible.
>
> **4. Bisection vs. ASTRA:** You are right that in a purely deterministic setting with exact gradients, bisection could theoretically be faster ($O(\log(1/\epsilon))$ vs $O(1/t)$), provided one fully solves the inner PGD problem at every step. The motivation for ASTRA is based on the *stochastic setting*: bisection is not straightforwardly applicable when gradients are stochastic, as we cannot get an exact error signal for the root-finding. ASTRA is designed as a stochastic approximation algorithm that handles the noise inherent in deep training.
>
> **5. Corollary 1 ($\delta_t = O(\beta_t)$):** The claim follows from the interaction between the contraction factor $\rho$ and the perturbation $\beta_t$. From Lemma 4, we have a recurrence of the form $\delta_{t+1} \le \rho \delta_t + C \beta_t$ with $\rho < 1$. Unrolling this yields $\delta_t \le \rho^t \delta_0 + C \sum_{i=0}^{t-1} \rho^{t-1-i} \beta_i$.
>     *   The term $\rho^t \delta_0$ decays exponentially (faster than any polynomial $\beta_t$).
>     *   The sum is a convolution of a geometric decay and the sequence $\beta_t$. For slowly decaying $\beta_t$ (e.g., polynomial decay), the sum behaves asymptotically as $O(\beta_t)$.
> So in the case of $\beta_t=0$, the result still holds (exponentially). The reviewer might have missed the $\rho<1$ factor.
>
> **6. Theorem 3 Proof (Locality and Circularity):**
> We appreciate the close reading of the convergence proof. You are correct that the result relies on the iterates remaining in the basin of attraction of $\lambda_*$. In our initialization, we start at $\lambda_0 = 0$. By design (Eq 14), $\lambda_*$ is the *minimal* stable regularization. Since the sparsity **gap $\Phi(\lambda)$ is positive for $\lambda < \lambda_*$**, the dynamics push $\lambda_t$ towards $\lambda_*$ without overshooting into instability, provided the step size $\beta_t$ is sufficiently small. The argument regarding $V_t = O(1/t)$ is not circular but inductive. We rely on classical results for stochastic approximation, which establishes that if $v_{t+1} \le (1 - \frac{A}{t})v_t + \frac{B}{t^2}$ and $A > 1$, then $v_t = O(1/t)$. The condition $2c\beta_0 > 1$ (mentioned in line 1058) is the standard condition ($A>1$) required to achieve the $O(1/t)$ rate. A detailed and more explicit reference is Francis Bach lecture on page 4: https://www.di.ens.fr/~fbach/orsay2016/lecture3.pdf

---

### Meta-Review · Area_Chair_J9MD · 2026-01-07

**Summary:**

The paper introduces an optimization-based framework for model compression and sparse training by adaptively tuning the sparse regularization parameter during training to satisfy hard sparsity constraints. While the reviewers found the goal of  a principled approach for solving its surrogate L1 regularized formulation given sparsity budget approach compelling, they raised significant concerns regarding technical writing and claims, and the lack of empirical evidence on full-scale foundation models as suggested by the title.

**Reviewer Concerns:**

Concerns regarding statistical significance in Table 1 were likely addressed in the rebuttal. However, two primary issues remain: the lack of empirical evidence on full-scale foundation models and various concerns regarding technical writing and theoretical claims. Specifically, Reviewer 9CbQ raised several points regarding the validity of the proofs. While the authors acknowledged certain errors and provided clarifications for some comments, these revisions are substantial enough to require another thorough review to ensure correctness. In addition, the other two reviewers noted that providing results only for ResNet-32 and a toy test on a single LLM head from Qwen is insufficient to substantiate claims of effectiveness for "Foundation Models." This concern remains unresolved.

**Reviewer Scores:**

It is difficult to project how Reviewer 9CbQ would have changed their score, as it depends on whether the author’s response successfully resolved their concerns; however, the reviewer may still feel that the overall technical writing requires significant improvement. The other two reviewers are likely to maintain their scores, as their primary criticisms regarding the limited scope of the experiments were not fully addressed in the rebuttal.

---

### Decision · Program_Chairs · 2026-01-26

Reject